# Design and directed evolution of noncanonical β-stereoselective metalloglycosidases

Woo Jae Jeong[1] & Woon Ju Song [1] ✉

Metallohydrolases are ubiquitous in nearly all subclasses of hydrolases, utilizing metal elements to activate a water molecule and facilitate its subsequent dissociation of diverse chemical bonds. However, such a catalytic role of metal ions is rarely found with glycosidases that hydrolyze the glycosidic bonds in sugars. Herein, we design metalloglycosidases by constructing a hydrolytically active Zn-binding site within a barrel-shaped outer membrane protein OmpF. Structure- and mechanism-based redesign and directed evolution have led to the emergence of Zn-dependent glycosidases with catalytic proficiency of $2.8 \times 10^9$ and high β-stereoselectivity. Biochemical characterizations suggest that the Zn-binding site constitutes a key catalytic motif along with at least one adjacent acidic residue. This work demonstrates that unprecedented metalloenzymes can be tailor-made, expanding the scope of inorganic reactivities in proteinaceous environments, resetting the structural and functional diversity of metalloenzymes, and providing the potential molecular basis of unidentified metallohydrolases and novel whole-cell biocatalysts.

Hydrolases are ubiquitous and indispensable in every living organism. They catalyze the cleavage of various substrates via the association of a water molecule. Depending on the types of hydrolyzed chemical bonds, hydrolases are classified into 13 enzyme commission numbers[1]. A large fraction of hydrolases are also categorized as metallohydrolases when metal elements are mechanistically involved with forming a nucleophile that reacts with the dedicated substrates[2].

Both metallo- and non-metallohydrolases have been discovered in nearly all subclasses of hydrolases, such as esterase, peptidase, and β-lactamase. However, glycosidase, which reacts with sugars, is an outstanding exception; very few glycosidases necessitate a metal ion[3], and even if they do, its role is limited to substrate binding[4,5] and stabilization of transition state[6] rather than to the direct activation of a water molecule and the cleavage of a glycosidic bond. Instead, canonical glycosidases primarily utilize a pair of acidic residues that function as nucleophile and/or general acid/base[7].

No discovery of metal-dependent glycosidase, namely metalloglycosidase, might be related to the intrinsic properties of metal elements; cationic metal ions may not accommodate a positively charged oxocarbenium ion-like transition state[8], although such

electrostatic repulsion perhaps can be overcome by the assistance of surrounding residues. In addition, inorganic complexes, metallopolymers, and metallopeptides showing glycosidase activities were reported[9–12]. This discrepancy led us to question whether metal-dependent glycosidases can be created within a biomacromolecular space. If so, the scope of artificial metalloenzymes and whole-cell biocatalysts can expand against the odds of natural emergence and selection[13,14].

Thus, we sought a versatile protein scaffold to build a metal-binding active site. We supposed that a protein containing a void space, such as β-barrel outer membrane protein F (OmpF) (Fig. 1a)[15], would be suitable for our study; its overall shape resembles the structure of host compounds, such as cyclodextrin (Fig. 1b), which may provide a structural basis for various functions, including catalysis, by encompassing guest molecules and forming noncovalent interactions[16,17]. In addition, OmpF is an outer membrane protein suitable for whole-cell catalysis and directed evolution.

In this work, we demonstrate that the structure and mechanism-guided protein redesign and directed evolution can transform OmpF into noncanonical Zn-dependent glycosidases with even high

[1]Department of Chemistry, Seoul National University, Seoul 08826, Republic of Korea. ✉e-mail: woonjusong@snu.ac.kr

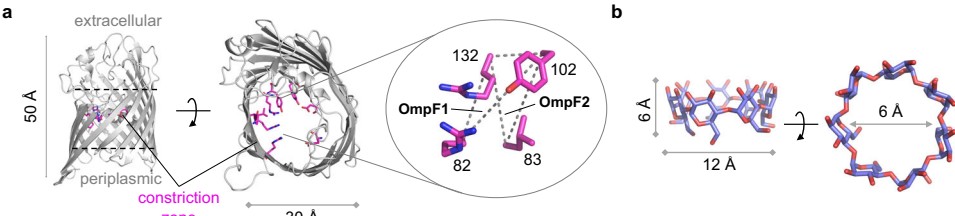

**Fig. 1 | OmpF as a versatile scaffold protein.** The structural resemblance of **a** OmpF (PDB 2OMF) and **b** cyclodextrin (CSD TEZZUV). Only one protomer of trimeric OmpF is shown for clarity. The horizontal dotted lines in **a** represent the location of lipid bilayers. The sets of three residues selected for installing putative Zn-binding motifs are depicted as dotted triangles.

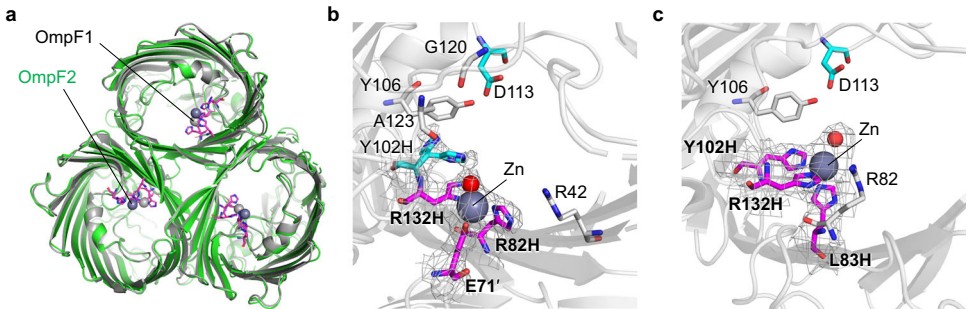

**Fig. 2 | X-ray crystal structures. a** The overlaid structures shown from the periplasmic side. The active sites of **b** OmpF1 and **c** OmpF2. Zn atoms and a metal-bound water molecule are shown as light navy and red spheres, respectively. Metal-ligating residues are colored in magenta and labeled in bold. Rationally redesigned and iteratively optimized residues are shown with cyan and gray sticks, respectively. The gray grid represents $2F_o - F_c$ electron density of the Zn-ligating site contoured at 1.0 σ.

β-stereoselectivity and catalytic activities under in vitro conditions and as whole-cell catalysts.

## Results and discussion

### Structure-based design of artificial Zn-binding proteins

We inspected the constriction zone of OmpF, the narrowest region, approximately halfway inside the barrel-shaped protein. A coordinatively unsaturated mononuclear Zn-binding site, a common catalytic motif in metallohydrolases, was created by placing three metal-coordinating histidines. R132H mutation was designed first because its side chain is oriented toward the eyelet and the local structure lies at the end of a β-strand; the latter may help to accommodate structural rigidity and flexibility for metal coordination. Then, we identified two positions whose the $C_\alpha$ atoms of residues are located within 3.8–10.6 Å of R132; the geometric boundaries are derived from the structural analysis of natural Zn-binding proteins (Supplementary Table 1). As a result, two triple mutants, R82H/Y102H/R132H (OmpF1) and L83H/Y102H/R132H (OmpF2), were designed as the parent templates.

### Structural characterization

Both OmpF variants were obtained from heterologous expression in *E. coli* as described previously[18]. After protein extraction, purification, and refolding, they were isolated as β-barrel trimers, retaining the native overall architecture of OmpF (Supplementary Fig. 3). It is also consistent with the X-ray single-crystal structures of OmpF1 and OmpF2 (Supplementary Table 3 and Fig. 2a). More importantly, both proteins possess a mononuclear Zn-binding site in the constriction zone. In OmpF1, two designed residues, R82H and R132H, coordinated to Zn ion, as expected (Fig. 2b). However, E71' from another monomer, instead of Y102H, was ligated to the Zn ion, resulting in a 2His/1Glu triad (2.0–2.4 Å for Zn-O/δN bonds) instead of a 3His triad. We later found that E71' is within the geometric range described above, and we dismissed that a latching loop from another protomer is accessible to the putative Zn-binding site. Regardless, the fourth site was ligated by a

non-proteinaceous molecule, tentatively assigned as a water molecule. Consequently, OmpF1 possesses a coordinatively unsaturated Zn-site, satisfying the prerequisite for metal-dependent hydrolysis. Due to the E71 binding, the Y102H mutation becomes no longer needed for Zn coordination, while the residue may still influence the chemical properties of the Zn-site. Therefore, we prepared another variant that reverses Y102H mutation (OmpF1Y) for further investigation.

The Zn-ligating site in OmpF2 comprised of three histidine residues (3His) in a tetrahedral geometry, resulting in 2.1–2.3 Å Zn-εN bonds (Fig. 2c). The fourth coordination site was possibly ligated by a water molecule, which is again suitable for our studies. Of note, the empty coordination site of OmpF2 pointed toward the periplasmic side, whereas that of OmpF1 is placed toward the extracellular region. Thus, they are likely to show distinct first-coordination spheres (2His/1Glu versus 3His), secondary coordination spheres, and drastically altered microenvironments, particularly when embedded in the outer membrane of *E. coli* cells.

### In vitro Zn-dependent hydrolytic activities of OmpF variants

To determine whether OmpF variants generate a Zn-mediated nucleophilic site for hydrolysis, we first measured esterase activities with a chromogenic substrate, *p*-nitrophenyl acetate (pNPA), with the purified proteins. Zn-dependent net activities were determined by observing the differences in the presence and absence of Zn ions (Fig. 3a, Supplementary Fig. 5, and Supplementary Table 5). The wild-type protein shows no detectable Zn-dependent hydrolytic activities. In contrast, all Zn-complexed OmpF variants exhibited considerably higher Zn-dependent esterase activities, demonstrating that they possess hydrolytically active Zn-binding sites, similar to synthetic, peptide-, and protein-based catalysts[19–21] and metalloesterases[2,22]. Their steady-state kinetic parameters showed the following order of OmpF1 < OmpF1Y ≤ OmpF2 and OmpF1 ≈ OmpF1Y ≤ OmpF2 for $k_{cat}$ and $k_{cat}/K_M$ values, respectively. These values are presumably determined by the combination of their discrete first-coordination spheres

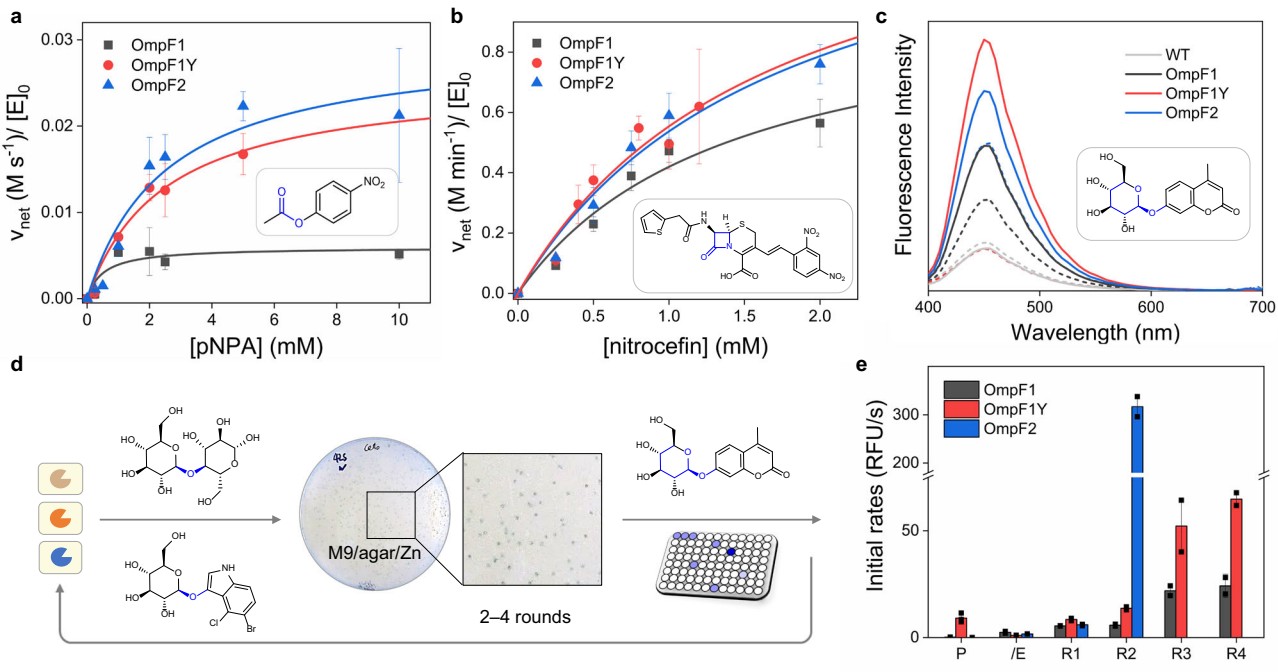

**Fig. 3 | Zn-dependent hydrolytic activities.** Kinetic analysis with **a** pNPA and **b** nitrocefin. **c** Fluorescence changes upon the hydrolysis of 4-β-MUG. Solid and dotted lines in **c** indicate the presence and absence of Zn ions, respectively. **d** Directed evolution of Zn-dependent β-glucosidase. **e** Whole-cell activity of the selected mutants on 4-β-MUG, where P and /E indicate the parent proteins (OmpF1, OmpF1Y, and OmpF2) and their D113E single-mutants, respectively. The data shown in **a** and **b** represent the averages and standard deviations of three independent experiments. The screening data shown in **e** represent the averages and standard deviations of two technical replicates. Source data are provided as a Source Data file.

(2His/1Glu in OmpF1 versus 3His in OmpF2), the directionality of Zn-bound water molecules, and surrounding microenvironments, including the residues at the 102 position (Y102H in OmpF1 versus Y102 in OmpF1Y).

They also showed Zn-dependent β-lactamase activities with nitrocefin in the order of OmpF1 < OmpF1Y ≈ OmpF2 (Fig. 3b, Supplementary Fig. 6, and Supplementary Table 5). Their catalytic efficiency or substrate-binding affinity were comparable to or higher than those of Zn-complexes[23,24] and artificial Zn-dependent β-lactamases[20]. In particular, all OmpF variants show saturation curves in Michaelis-Menten analysis, suggesting the OmpF protein scaffold may have a substantial binding affinity with the β-lactam analog even before any sequence optimization. It might be related to the native role of OmpF to be the native passage of antibiotics into the cells[25], implicating that OmpF is a versatile scaffold that functions as a host-like macromolecule and interacts with guest-like molecules for catalysis.

Finally, we determined whether Zn-complexed OmpF variants are catalytically competent in the hydrolysis of glycosides. From this point, we modified the expression vectors to translocate OmpF variants to the outer membrane of *E. coli* cells to prepare folded membrane proteins directly (Supplementary Fig. 7). The extracted OmpF variants exhibited fluorescence increase upon the addition of a fluorogenic substrate, 4-methylumbelliferyl-β-D-glucopyranoside (4-β-MUG), only at Zn-bound states (Fig. 3c). The formation of the hydrolyzed product, 4-methylumbelliferone, was also detected by HPLC (Supplementary Fig. 8), demonstrating that OmpF variants indeed hydrolyze the glycosidic bond of 4-β-MUG as Zn-dependent glycosidases.

### Structure- and mechanism-based redesign of OmpF variants
The canonical Koshland mechanism suggests that inverting and retaining β-glycosidases operate via a pair of acidic residues that are 6–11 Å and 5.1–5.5 Å apart, respectively[7,26,27]. We surmised that the

analogous mechanism is operative to OmpF variants. In that case, the glycosidase activity might arise from assisting at least one pre-existing acidic residue near the Zn-site (Supplementary Fig. 4 and Supplementary Table 4). In particular, D113 shows proper orientation and interatomic distance between the terminal oxygen atom and Zn-bound exchangeable ligands as 7.9–9.5 Å and 4.7–5.1 Å in OmpF1 and OmpF2, respectively (Fig. 2b, c). In addition, D113E mutation (denoted as /E) altered the glycosidase activities of OmpF variants; OmpF1/E and OmpF2/E showed substantially elevated activities, whereas that of OmpF1Y/E was somewhat reduced (Supplementary Fig. 8). Although the impacts of D113E mutation are dissimilar with OmpF variants, these data suggest that the acidic residue is functionally coupled with the Zn-site, possibly playing an essential role in glycosidase activities.

Of note, OmpF1/E, but not others (OmpF1, OmpF1Y, OmpF2, OmpF1Y/E, and OmpF2/E), exhibited glycosidase activity even as the Zn-free apo-state (Supplementary Fig. 8), implying that alternative catalytic pairs, possibly two acidic residues, might have co-emerged serendipitously along with Zn-mediated ones. Because wild-type protein or D113E single mutant showed no such activity with 4-β-MUG, the unexpected activities of OmpF1/E at the apo-states are likely to be associated with the introduction of mutations for Zn-ligation (R82H or R132H). Regardless, all OmpF variants are more active as Zn-bound forms than in the apo-states, indicating that they still primarily function as Zn-dependent glycosidases.

### Optimization of the active site pockets by directed evolution
With three D113E mutants (OmpF1/E, OmpF1Y/E, and OmpF2/E) as initial templates, we iteratively constructed whole-cell mutant libraries. Then, they were screened with two substrates with β−1,4-glycosidic linkages, cellobiose and 5-bromo-4-chloro-3-indolyl-β-D-glucopyranoside (X-Glu), simultaneously in Zn-supplemented growth medium (Fig. 3d). The growth and formation of blue-colored colonies indicated that the glucose and 5,5′-dibromo-4,4′-dichloro-indigo are produced

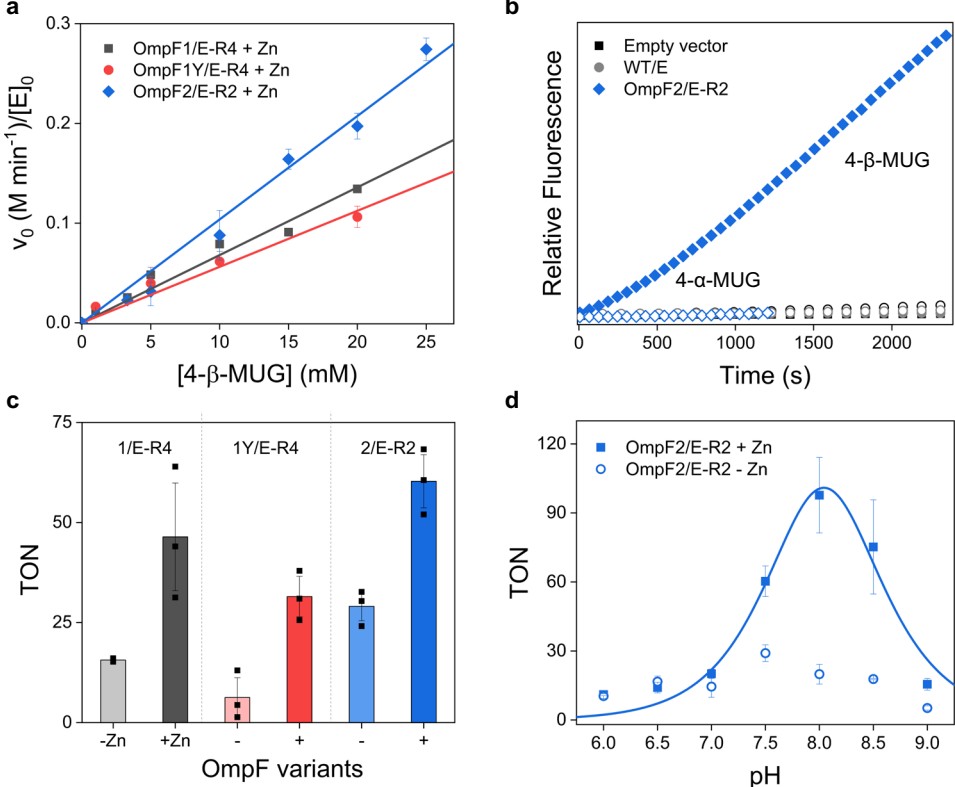

**Fig. 4 | Glucosidase activities of OmpF variants with 4-β-MUG. a** Kinetic analysis. **b** Whole-cell activities of OmpF2/E-R2. **c** Turnover number (TON) in the absence and presence of Zn ions. **d** The pH-dependent activities of OmpF2/E-R2. The data shown in **a**, **c**, and **d** represent the average and standard deviation of three independent experiments. Source data are provided as a Source Data file.

sufficiently as a carbon source for cell growth and an insoluble blue pigment, respectively. The catalytically active cells were further sorted by the reactivities with 4-β-MUG. As a result, the following mutants were obtained: R42S/Y106A/G120C/A123N (OmpF1/E-R4), Y106R/R42E/G120S/A123V (OmpF1Y/E-R4), and Y106H/R82C (OmpF2/E-R2) (Fig. 3e). They showed significantly elevated whole-cell activities with the initial rates up to two orders of magnitude relative to those of their parent proteins (P), OmpF1, OmpF1Y, and OmpF2, respectively.

### Characterization of the evolved OmpF variants

For more accurate kinetic studies, we measured the glycosidase activities of the evolved variants under in vitro conditions. They are active in the order of OmpF1Y/E-R4 < OmpF1/E-R4 < OmpF2/E-R2 with the second-order rate constants, $k_2 = 5.6$–$10.4$ min$^{-1}$ M$^{-1}$ (Fig. 4a). The kinetic parameters account for up to $2.8 \times 10^9$-fold enhancement from the uncatalyzed rate[28] (Supplementary Fig. 12 and Supplementary Table 6). Intriguingly, the iterative mutations also elevated the glycosidase activities of their apo-forms, suggesting that the alternative Zn-independent reaction routes have developed simultaneously via sequence optimization. However, the inductively coupled plasma-mass spectrometry (ICP-MS) analysis demonstrated that OmpF2/E-R2 protein shows a Zn-bound state on the outer membrane of E. coli cells (Supplementary Table 7), indicating that whole-cell activities are primarily derived from the coordinatively unsaturated Zn-sites.

When 4-α-MUG with an α–1,4-glycosidic linkage was employed instead of 4-β-MUG, all evolved OmpF variants forms displayed nearly no or negligible whole-cell activity (Fig. 4b and Supplementary Fig. 11). Such high β-stereoselectivity is likely to be related to the structures of three substrates, cellobiose, X-Glu, and 4-β-MUG, used for iterative sequence optimization. Thus, their chiral active site pockets might have been optimized for β-glycosides via the orchestrated interactions with the adjacent residues such that 4-β-MUG,

but not 4-α-MUG, can be positioned as a catalytically relevant orientation.

We also measured the turnover number (TON) of the evolved variants with 4-β-MUG, the mole of products per that of enzyme, using cell lysates (Fig. 4c). The values were increased up to ~100-fold relative to those of the parents, again reflecting the chemical power of sequence optimization. The removal of the Zn ion or the mutation of adjacently located acidic residues, such as E113, E117, and E62 (Supplementary Fig. 13), partially or entirely inactivated the enzymes, suggesting that a Zn-site and at least one acidic residue in proximity constitute a noncanonical catalytic motif of Zn-dependent glycosidases.

In addition, we observed that the OmpF2/E-R2 variant could hydrolyze the β-glycosidic bond in n-octyl-β-D-glucopyranoside (OG), which was initially added as a nonionic detergent for the preparation of membrane proteins. The formation of glucose and 1-octanol was detected as the hydrolyzed products (Supplementary Fig. 14), indicating that the evolved OmpF variant accommodates a hydrophobic and bulky substrate that was not even used for selection.

All three evolved OmpF variants showed pH-dependent activities with 4-β-MUG (Fig. 4d and Supplementary Fig. 15). Both Zn-complexed and apo-states show bell-shaped pH-dependence, revealing at least two ionizable side-chains to be essential for catalysis (p$K_a$ = ~7.7 and 8.3). It is consistent with natural glycosidases having two discrete p$K_a$ values[29,30]. However, Zn-complexed and the apo-state of OmpF variants yielded the maximal TONs at pH 8.0–8.5 and 7.5–8.0, respectively, whereas most natural glycosidases show the pH optimum at pH 5.0–6.5[30–32]. But glycosidases, such as alkaline xylanases, exhibit catalytic reactivity at more basic conditions[33–35]. The pH-dependence of OmpF variants might be derived from replacing one of an acidic pair with a Zn-OH$_2$/OH moiety. Besides, the unique chemical environments of OmpF might account for the pH-dependence; it was reported that

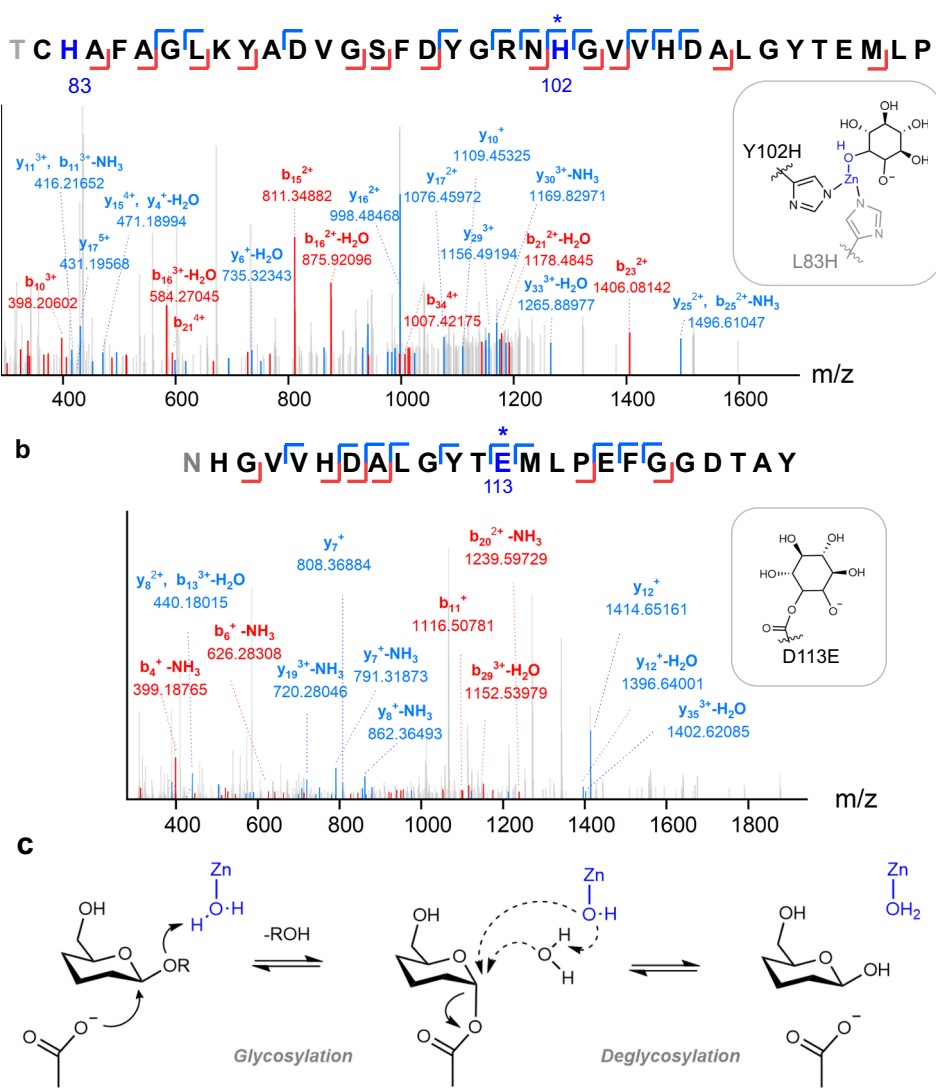

**Fig. 5 | Mechanistic studies of OmpF2/E-R2 variant.** The representative tandem LC/MS mass fragmentations after treating with CBE inhibitor with **a** Zn-bound and **b** apo proteins. Only major peaks are labeled for clarity. The proposed structures of Zn(OH)-CBE ligated to Y102H and L83H and CBE-conjugated D113E are shown as insets. **c** A proposed mechanism of Zn-dependent retaining β-glycosidase. The association of a water molecule is omitted for clarity.

the $pK_a$ values of the side-chains in the constriction zone are considerably perturbed from those of amino acids in bulk solvents[36–38].

To further identify the catalytic motifs of OmpF variants, a mechanism-based covalent inhibitor, conduritol B epoxide (CBE)[39,40], was added, and the resulting proteins were analyzed by trypsin digestion and tandem LC/MS spectrometry. CBE molecules are bound to at least two positions of Zn-complexed OmpF variants, one to the Zn(OH) complex ligated to either R82H or Y102H residues (Fig. 5a) and the other to the adjacently located acidic residues, such as E62, D113E, and E117 (Supplementary Fig. 16 and Supplementary Table 8). In the apo-states, CBE was conjugated to D113E (Fig. 5b) or additional positions, such as D92 or E117. These results contrast with the wild-type protein and D113E single mutant showing no attachment of CBE in the constriction zone.

We also analyzed the reaction product with 4-β-MUG to identify whether the retaining versus inverting mechanism is operative. Because glucose shows too rapid mutarotation, we conducted activity assays in the presence of excess azide, which functions as an external nucleophile, as described previously[30,41,42] (Supplementary Fig. 17). Although OmpF2/E-R2 still yielded glucose exclusively, OmpF2/E-R2* variant (Y106H/R82Y), which differs from OmpF2/E-R2 only by a single residue at the position 82, produced the mixtures of glucose and

1-azido-1-deoxy-glucose. The ¹H NMR analysis demonstrated that the hydrolyzed product is β-glycoside, indicating that we created a retaining β-glycosidase that operates via two sequential reactions, glycosylation and deglycosylation, from the actions of two catalytic motifs as a nucleophile and a Lewis acid/base.

Therefore, we proposed a reaction mechanism of Zn-dependent glycosidases by revising the Koshland mechanism, where one of the canonical acid pairs is replaced by a coordinatively unsaturated Zn-site (Fig. 5c). For retaining β-glycosidases, such as OmpF2/E-R2, an acidic residue, possibly D113E, may initiate glycosylation as a nucleophile when $Zn-OH_2$ species acts as a Lewis acid. Then, the resulting Zn-OH may become a direct nucleophile or activate a water molecule as a Lewis base, facilitating the deglycosylation step.

Docking simulations with Zn-complexed OmpF1/E-R4 and OmpF2/E-R2 model structures also suggest that the β-glycosidic bond of 4-β-MUG may be positioned at the appropriate orientation to the coordinatively unsaturated Zn-site and D113E in the constriction zone as described above (Supplementary Fig. 18). Therefore, our work demonstrates that coordinatively unsaturated Zn-site can directly mediate the hydrolysis of glycosidic bonds, overcoming potential energy barriers with the formation of an oxocarbenium ion-like transition state during glycoside hydrolysis.

The structure- and mechanism-based design of artificial metalloenzymes allows us to validate our level of understanding of the chemical interplay between metal elements and protein environments. This study demonstrated that the retrosynthetic construction of the active sites, reaction mechanism-based redesign, and directed evolution could create artificial metalloglycosidases, enabling inorganic cofactors to activate glycosidic bonds in protein environments. The successful conversion of neither catalytic nor metal-binding membrane protein OmpF into various metallohydrolases indicated that OmpF is apt for introducing an active site, reactions with biologically or non-biologically relevant substrates, and even whole-cell catalysis. Although membrane proteins have rarely been adapted for enzyme designs, our work demonstrated that OmpF could be a host macromolecule for various potential applications. Then, it leads to an intriguing question, why is there no precedence for such metal-dependent glycosidases in Nature? It might be related to the alternative metalloenzymes and relevant biomimetic complexes[43–45], which react with glycosides via Cu-dependent oxidative cleavage mechanism. Alternatively, there might still be unidentified metalloproteomes and metalloenzymes. Along this line, it is worthy to note that a novel glycosidase has been discovered and recently characterized, where Zn-coordinating cysteine functions as a direct nucleophile for glycosidase via a retaining mechanism[46]. They suggested that the Zn ion plays a direct role in tuning the nucleophilicity of the catalytic cysteine and the energetics of metalloenzymes, broadening the role of metallocofactors in glycosidases. In addition, the current study demonstrated that Zn-dependent and independent catalytic routes can co-emerge serendipitously. Such promiscuous reactivity might be related to divergent evolution[47]. If then, artificially designed enzymes may resemble the states of primordial or ancestor enzymes[48].

## Methods

### Structural analysis of OmpF

To create a coordinatively unsaturated mononuclear Zn-binding site on a protein scaffold, we manually curated and inspected the crystal structures of the proteins that natively possess a Zn-binding motif. The distance between the $C_\alpha$ atoms of the ligating residues was measured using the PyMOL program (Fig. 1, Supplementary Fig. 1, and Supplementary Table 1). The metal-ligating residues are located within 3.8–10.6 Å between their backbone $C_\alpha$ atoms, regardless of their sequence, overall structure, and function. Therefore, the values were applied as geometric restraints for our structure-guided design of the Zn-binding sites.

### Design of triad mutants

The constriction zone was inspected to install a Zn-binding motif (Fig. 1 and Supplementary Fig. 2). We first narrowed down the candidate positions into the following: Y102, Y106, D113, E117, and R132, due to the orientation of their side-chains, which pointed towards the void space of the cylindrical protein. The residues were prioritized depending on their secondary structures and local flexibility. R132 was the most desirable site as the first vertex of a Zn-binding triad, and the other two vertexes were determined based on the optimal range of the interatomic $C_\alpha$ distance, which was obtained as described above. As a result, two sets of triads, OmpF1 and OmpF2, comprising R132H/Y102H/R82H and R132H/Y102H/L83H, respectively, were determined.

### Sample preparation

A plasmid responsible for OmpF heterologous expression (pET28b/kan$^R$/ompf) was a generous gift from Professor Seokhee Kim at Seoul National University. We deleted a His-tag and a thrombin cut-site prior to the N-terminus using polymerase chain reaction (PCR) with custom-designed primers (Supplementary Table 2a). The PCR products were transformed into *E. coli* DH5α competent cells for isolation and sequencing (pET28b/kan$^R$/ΔHis/ompf).

For the preparation of OmpF triad mutants, site-directed mutagenesis was carried out using custom-designed primers. After digestion with DpnI restriction enzyme for 1.5 h at 37 °C, the plasmids were transformed into *E. coli* DH5α competent cells. Each colony was inoculated in 5 mL LB media with 50 mg/L kanamycin at 37 °C, and the plasmid was extracted for sequencing (Macrogen or Bionics).

OmpF variants were prepared as reported previously[18]. In short, the plasmids were transformed into *E. coli* BL21(DE3) competent cells. A few colonies were inoculated in LB medium with 50 mg/L kanamycin. After overnight growth at 37 °C, the culture was inoculated in LB medium containing 50 mg/L kanamycin at 37 °C in an orbital shaker. After the optical density of the culture ($OD_{600}$) reached 0.7, protein expression was induced by the addition of isopropyl β-D-1-thiogalactopyranoside (IPTG) at a final concentration of 1 mM at 37 °C for 4 h. The cells were harvested via centrifugation at 5000 rpm ($4715 \times g$) and 4 °C for 15 min, and the pellets were stored at −80 °C for further use.

The cells were thawed in 50 mM Tris/HCl (pH 8.0) buffer and subjected to lysis via sonication. The lysates were centrifuged at 13,000 rpm ($18,800 \times g$) and 4 °C for 30 min, and the pellets were washed with 1% (v/v) Triton X-100 in 50 mM Tris-HCl (pH 8.0) buffer. The buffer with 8 M urea was added to the pellet and incubated for 4 h at 37 °C. The inclusion bodies were removed via centrifugation at 13,000 rpm ($18,800 \times g$). The resulting supernatant was loaded on a Q anionic exchange column (HiTrap Q HP, GE Healthcare Life Sciences) using a protein purification system (ÄKTA Prime Plus). The OmpF protein was eluted by applying a linear gradient of 1 M NaCl, and the purity of the proteins was analyzed using 15% SDS-PAGE (Bio-rad). The pure fractions were concentrated at 4 °C using a 30 kDa cutoff centrifugal filter (Amicon) or a stirred cell system (EMD, Millipore) with a 10 kDa cutoff membrane. The protein concentration was determined using a UV/vis spectrophotometer (Agilent, Cary 8454) using the extinction coefficients at 280 nm estimated from the protein sequence.

The detergents used for protein refolding were purchased from Anatrace, Thermofisher, Avanti Polar Lipids, and GoldBio. In short, nonionic detergents, 1% (w/v) n-dodecyl-β-D-glucopyranoside and 1% (w/v) n-dodecyl-β-D-maltopyranoside were added to the unfolded OmpF protein (20 mg/mL), followed by the incubation at 37 °C for 1–2 days. The protein folding and oligomerization were determined using 15% SDS-PAGE analysis (Supplementary Fig. 3c). The refolding efficiency was measured to be ~80%. The unfolded proteins (around 20%) were removed via trypsin digestion (Sigma−Aldrich, 0.1 mg/mL) overnight at 37 °C, followed by anion exchange column chromatography using a 50 mM Tris/HCl (pH 8.0) buffer containing 0.5% OG and NaCl (0–1 M). Metal-free protein samples were prepared by adding 5-fold molar excess of ethylenediaminetetraacetic acid (EDTA) and incubating overnight at 4 °C. The excess EDTA was removed by washing with a metal-free 50 mM Tris/HCl (pH 8.0) buffer containing 0.5% OG several times, using a 30 kDa cutoff centrifugal filter.

### Protein crystallization, structure determination, and refinement

Prior to crystallization, the protein samples were washed with a 50 mM Tris/HCl (pH 8.0) buffer containing 0.5% OG and 100 mM NaCl. ZnCl$_2$ solution was added to the resulting protein at the ratio of 1.2 equiv. of Zn ion to the protomer. The samples were used for crystallization using a sitting-drop method at 20 °C, and the reservoir solution was a 0.1 mM sodium cacodylate (pH 8.0) buffer containing 43% PEG 200 and 0.12 M MgCl$_2$ as described previously[49]. The detailed information about crystallization, data collection, and refinement statistics are summarized in Supplementary Table 3. The diffraction data were collected in 7 A beamline at Pohang Accelerator Laboratory (PAL).

X-ray diffraction data were processed and scaled using the program suite HKL2000[50] and XDS[51]. All data were processed using CCP4, including Pointless, Matthews, and Scala5[52]. Molecular replacement was performed using either Molrep or Phaser[53,54] using the structure of

the wild-type OmpF (PDB 2OMF or 2ZFG) as a search model. Rigid-body and restrained refinement using REFMAC5 and Phenix[55], along with manual model rebuilding and COOT[56], were used to obtain the structural models. Non-crystallographic symmetry restraints were also applied throughout the refinement[57]. Residues or side-chains with low electron density and B-factor (>100 Å$^2$) were omitted. All the structural figures were produced using PyMOL. The geometric parameters of the Zn-binding sites in the OmpF1 and OmpF2 variants are summarized in Supplementary Table 4.

### In vitro esterase and β-lactamase activity assays

Chromogenic substrates, such as *p*-nitrophenyl acetate (pNPA) and nitrocefin, were purchased from Alfa Aesar and Cayman, respectively. The refolded OmpF variants at a final concentration of 10 μM in 50 mM Tris-HCl (pH 8) buffer were pre-mixed with 1.2 equiv. of Zn ions (ZnCl$_2$) to OmpF monomer on a 96-well plate. The reaction solution was prepared by mixing 90 μL of protein solution with 10 μL of the substrates dissolved in dimethyl sulfoxide (DMSO). The time-dependent optical changes were monitored at room temperature upon the addition of various concentrations of pNPA or nitrocefin at 410 or 486 nm, respectively, using a microplate reader (Biotek Synergy H1m). The kinetic data were collected using BioTek Gen5 software version 3.02 and analyzed with Origin 2021 (64-bit) 9.8.0.200 (academic). The net esterase activity of the Zn-dependent OmpF variants was determined by detecting the differences in the product formation rates in the presence and absence of Zn ions, in which free Zn ion exhibits no detectible activity (Supplementary Fig. 5). The net β-lactamase activity of the Zn-dependent OmpF variants was determined by detecting the differences in the product formation rates in the presence and absence of the OmpF variants, in which the apo-protein exhibits no detectible activity (Supplementary Fig. 6). The steady-state kinetic parameters were obtained via an iterative nonlinear fit to a Michaelis-Menten equation or a linear fit using Origin 2021 (64 bit) 9.8.0.200 (academic).

### Expression and extraction of the OmpF variants in a folded form

To bypass the protein refolding process, we expressed and extracted the OmpF variants in a folded form from the outer membrane of *E. coli*, as described previously[58]. For the translocation of the OmpF variants to the outer membrane, the native signal peptide (22 amino acids, MMKRNILAVIVPALLVAGTANA) was placed prior to the N-terminus of OmpF, resulting in plasmid, pET28b/kan$^R$/sig/ompf (Supplementary Fig. 7). For protein expression, *E. coli* BL21 (DE3) competent cells lacking LamB/OmpC/OmpF (ΔBCF, #102269 from Addgene)[59] were used. After overnight growth in LB medium, the cells were inoculated and grown in low salt LB medium with 50 mg/L kanamycin at 37 °C in an orbital shaker at 150 rpm until the optical density (OD$_{600}$) reached 0.6. After the addition of IPTG at a final concentration of 0.1 mM and overnight incubation at 25 °C, the cell pellets were harvested via centrifugation at 5000 rpm and 4 °C. Cell lysates were obtained via homogenization with a 50 mM Tris/HCl (pH 8.0) buffer, followed by centrifugation at 13,000 rpm (18,800 × $g$) and 4 °C for 30 min. The pellets were resuspended and incubated with a 50 mM Tris/HCl (pH 8.0) buffer containing 2% (v/v) Triton X-100 for 2 h at 25 °C. The solution was ultra-centrifuged (Optima L-100K) at 40,000 rpm (164,400 × $g$) for 1 h. The inner membrane fractions were removed by decanting the supernatants, and the remaining outer membrane fractions were incubated with a 50 mM Tris-HCl (pH 8.0) buffer containing 3% (w/v) OG at 25 °C overnight. After ultracentrifugation, the supernatant was treated with 0.1 mg/mL trypsin at 37 °C overnight, if necessary. The folding and oligomeric states of the proteins were analyzed using 15% SDS-PAGE analysis.

### Initial glycosidase activity assays

As a fluorogenic substrate, 4-β-MUG (GoldBio) was added to the OmpF variants extracted directly from the cells (Fig. 3c and Supplementary Fig. 8) when proteins (5–10 μM) in 50 mM Tris/HCl (pH 7.5) buffer

containing 3% OG were pre-mixed with Zn ion (ZnCl$_2$, 20 μM) or the excess EDTA (1 mM). The excitation and emission wavelength were set as 370 and 450 nm, respectively, with a gain of 50 or 100. In addition, the consumption of 4-β-MUG and the subsequent formation of 4-methylumbelliferone was detected by HPLC (Agilent 1220 Infinity II), using Agilent ChemStation software version C.01.09. The reaction mixture (10 μL) was injected into an InfinityLab Proshell column (120 EC-C18 4.6 × 100 nm or 120 EC-C18 4.6 × 150 mm, 2.7 μm; particle size: 2.7 μm). A linear gradient of H$_2$O containing 0.05% trifluoroacetic acid (TFA) and CH$_3$CN containing 0.1% TFA solvents, from 9:1 to 0:10, was applied for 25 min, and the elution was monitored at 300 nm (Supplementary Fig. 8d).

### Construction of mutant libraries

For sequence optimization, we selected two or four residues that are located in the vicinity of the Zn-binding sites: the R42, Y106, G120, and A123 residues of OmpF1/E; the Y106, R42, G120, and A123 residues of OmpF1Y/E; and Y106 and R82 residues of OmpF2/E. The mutant libraries were constructed via saturation mutagenesis using custom-designed primers containing NDT and VHG codons (N = A/G/C/T; D = A/G/T; V = A/C/G; H = A/C/T) for the selected sites, as shown previously[60](Supplementary Table 2b). A combination of the NDT and VNK codons, instead of NNK, was used to exclude the stop codon and tryptophan. After transformation of the PCR products into *E. coli* DH5α competent cells, more than 100 colonies were pooled to generate a single-site randomized library with a 95% confidence level, as described previously[61].

### Screening of mutant libraries

The mixture of plasmids containing a native signal peptide (pET28b/kan$^R$/sig/ompf) was transformed into *E. coli* BL21(DE3) ΔF (#102259, Addgene) and were grown in LB/agar plates containing 50 mg/L kanamycin at 37 °C. The cultures were grown overnight in LB broth supplemented with kanamycin (50 mg/L) at 37 °C in an orbital shaker at 150 rpm, and were inoculated in 100 mL of medium and grown until the OD$_{600}$ reached 0.6. OmpF expression was induced by the addition of 0.1 mM IPTG and 50 μM ZnCl$_2$, followed by incubation for 16 h at 25 °C. The cultures were spread on a M9/Agar plate containing 50 mg/L kanamycin, 0.2–0.4% cellobiose (Supplementary Fig. 9a), and 20 mg/L X-Glu (GoldBio) (Supplementary Fig. 9b), and inoculated for 1–2 days at 25 °C. Blue colonies (Supplementary Fig. 9c) were selected and inoculated in 600 μL of LB medium containing 50 mg/L kanamycin on a 96 deep-well plate and grown at 37 °C and 290 rpm (N-Biotek). After overnight incubation at 37 °C, 540 μL of each culture was stored for DNA sequencing. The rest of the cultures were diluted with 540 μL of fresh LB and induced with 0.1 mM IPTG and 50 μM ZnCl$_2$ for 16 h at 25 °C, until the OD$_{600}$ reached 0.6. The catalytic activity for 4-β-MUG was measured at excitation and emission wavelengths of 370 and 450 nm, respectively, using a microplate reader (Biotek Synergy H1m). The measurements were carried out in duplicates. The colonies showing higher activity values than those obtained in the preceding round were regrown for isolating plasmid, sequencing, and further activity assays (Supplementary Fig. 10).

### Whole-cell activity assays

The colonies from each round of screening showing the highest glycosidase activity were transformed into *E. coli* BL21(DE3) ΔBCF competent cells (#102269). Colonies were incubated in LB broth supplemented with kanamycin (50 mg/L) at 37 °C for cell growth. The expression of the mutants was induced using 0.1 mM IPTG and 50 μM ZnCl$_2$ for 16 h at 25 °C until an OD$_{600}$ of 0.6 was obtained. After protein expression, the whole-cell activities were measured by mixing 45 μL of whole cells with 5 μL of the substrates (4-α-MUG or 4-β-MUG at 10 mM dissolved in DMSO) (Fig. 4b and Supplementary Figs. 10, 11). Time-dependent fluorescence changes were monitored by using a

microplate reader and a 384-well black plate (Corning 3575). The fluorescence changes were monitored at 450 nm upon excitation at 370 nm with a focal height of 8.5 mm and a gain of 50 or 100. The concentration values of the OmpF variants were quantified using densitometric analysis with SDS gels, where the protein samples were extracted from 100 mL of whole cells using 1 mL of Tris/HCl (pH 8.0) buffer containing 3% OG. The normalized activity was determined by dividing the observed whole-cell activity by the protein concentration of the cell lysates (Supplementary Fig. 10).

### Kinetic analysis of the evolved glycosidases under in vitro conditions

For kinetic analysis, the proteins extracted from whole cells were further purified using an anion exchange column chromatography (HiTrap Q HP, GE Healthcare Life Sciences) with 50 mM Tris/HCl (pH 8.0) buffer containing 0.5% OG using a linear gradient of NaCl (0–1 M). For the preparation of a metal-free protein sample, a 5-fold molar excess of EDTA was added and incubated overnight 4 °C. The excess EDTA was removed using a 30 kDa cutoff centrifugal filter and a metal-free buffer. The reaction mixture was prepared by mixing 45 μL of protein pre-incubated with 1.2 equiv. of Zn ion ($ZnCl_2$) and 5 μL of 4-β-MUG (10 mM stock dissolved in DMSO).

Time-dependent fluorescence changes were monitored upon the addition of 4-β-MUG at various concentrations (0–20 mM) using a microplate reader (Biotek Synergy H1m) and a 384-well black plate (Corning 3575). The excitation and emission wavelengths were 370 and 450 nm, respectively, with a focal height of 8.5 mm and a gain of 100. Time-dependent fluorescence changes were measured and fit to a linear function to obtain the initial molar rates (Fig. 4a and Supplementary Table 6).

The rate constant of uncatalyzed hydrolysis ($k_{uncat}$) was measured by mixing 20 μL of 4-β-MUG (10 mM stock dissolved in DMSO) with 180 μL of 50 mM Tris (pH 7.5) buffer containing 3% OG at various temperature values (55–75 °C) using a thermocycler (Bio-rad). Every 24 h, the formation of 4-methylumbelliferone was quantified using a microplate reader at excitation and emission wavelengths of 370 and 450 nm, respectively, with a gain of 58. The $k_{uncat}$ value at 25 °C was estimated by a linear fit to the Arrhenius equation.

### ICP-MS analysis

The metal element and quantity of OmpF2/E-R2 were determined by ICP-MS. For sample preparation, the protein was extracted from whole cells and purified using size exclusion chromatography (Superdex 200 10/300 GL) using 50 mM Tris-HCl (pH 8.0) buffer pre-treated with Chelex 100.

### TON measurements

The turnover number (TON) was determined by mixing cell lysates (450 μL) containing 5–10 μM of the OmpF variants and 4-β-MUG (50 μL of 10 mM stock dissolved in DMSO). For the preparation of Zn-complexed and apo proteins, $ZnCl_2$ and EDTA were added prior to the addition of 4-β-MUG. After incubation for 16 h, the concentration of 4-methylumbelliferone as the hydrolyzed product was quantified by using fluorescence and HPLC as described above. For fluorescence, the excitation and emission wavelength were set as 370 and 450 nm, respectively, with a gain of 50 or 100 (Fig. 4e and Supplementary Fig. 14).

For the reactions with OG, Zn-complexed OmpF2/E-R2 was reacted for 48 h, and the solution was analyzed by HPLC and GC-MS (Trace 1310/ISQ LT single quadrupole mass spectrometer, Thermo Scientific) to detect the formation of glucose and 1-octanol, respectively. The GC-MS data were collected using Xcalibur software version 4.0.0.29. For pH-dependent assays, TONs were measured with 4-β-MUG by using the following buffers; 50 mM Bis-Tris (pH 6.0–6.5) and 50 mM Tris (pH 7.0–9.0).

### Inhibitor assays

CBE (Carbosynth)[39,40] at a final concentration of 500 μM in ddH$_2$O was added to the extracted proteins at the final concentration of 10 μM. After incubation for 4 h at 25 °C, 50 μL of 4-β-MUG (10 mM stock solution in DMSO) was added to the cell lysates (450 μL), and TON values were analyzed as described above.

### Tandem LC/MS analysis

The extracted proteins were pre-mixed with or without Zn ions and treated overnight with the covalent inhibitor CBE at a 200-fold molar ratio to the proteins. The samples were heated at 70 °C for 1.5 h using a thermocycler and treated with trypsin at 37 °C overnight. After the SDS-PAGE experiment, the bands corresponding to the target proteins were excised and treated with DTT (10 mM) and iodoacetamide at the final concentrations of 10 mM and 55 mM, respectively. After digestion with trypsin/chymotrypsin (Thermo Fisher Scientific) and desalting using ZipTip C18 (Millipore), the resulting peptides were resuspended in 0.1% (v/v) formic acid (Merck) and analyzed using LC-MS/MS (Q Exactive Hybrid Quadrupole-Orbitrap, Thermo Fisher Scientific) and Acclaim PepMap 100 trap columns (100 μm × 2 cm, nanoViper C18, 5 μm, 100 Å or 75 μm × 15 cm, nanoViper C18, 3 μm, 100 Å), as published previously[62]. The injected samples were washed for 6 min with 98% solvent A (water and CH$_3$CN (98:2 v/v) and 0.1% formic acid) at a flow rate of 4 μL/min or 300 nL/min, respectively using the following elution program; a linear gradient of 2–40% solvent B for 45 min, a gradient of 40–95% for 5 min, elution with 95% solvent B (100% CH$_3$CN and 0.1% formic acid) for 10 min, and 2% solvent B for 20 min. The raw data were collected using Xcalibur software version 4.3. Subsequently, the raw data were processed using Proteome Discoverer 2.3 (Thermo Fisher Scientific) using the sequence of OmpF (UniProt P02931) and the OmpF variants. Methionine oxidation, cysteine carbamidomethylation, and the covalent linkage of a CBE molecule to glutamate, aspartate, or Zn-hydroxide bound to 3His or a 2His/1Glu site were included as specified modifications. A strict (1%) and a relaxed (5%) protein false discovery rates were applied using peptide spectrum matches validator node in Proteome Discoverer (Supplementary Fig. 16 and Supplementary Table 8). Of note, only the fragments containing Zn-CBE moiety exhibited the distribution of natural zinc isotopes (Supplementary Fig. 16).

### Characterization of evolved glycosidases

Activity assays were performed with OmpF2/E-R1* variant, which showed 14.3% of TON relative to OmpF2/E-R2, in the presence of sodium azide[29,63]. After mixing 800 μL of 20 μM OmpF variants, 100 μL of 100 mM 4-β-MUG dissolved in DMSO, and 100 μL of 1 M sodium azide (Supplementary Fig. 17a), the formation of glycosyl azide was detected using LC-MS (Agilent Technologies 1200 Infinity Series equipped with Agilent Technologies 6120 Quadrupole LC/MS) using Agilent ChemStation software version B.04.03.

For structural characterization, glycosyl azide was purified by injecting the reaction mixture (4 mL in total) into an HPLC ZORBAX SB-C18 Semi-Preparative 9.4 × 250 mm, 5 μm column in a HPLC system (Agilent 1260 Infinity II). A linear gradient of two solvents, H$_2$O containing 0.05% TFA and CH$_3$CN containing 0.1% TFA, was applied at a ratio of 9:1 to 0:10 over 30 min at 2 mL/min. The eluted glycosyl azide was freeze-dried and subsequently reacted with Cu(I) (0.01 equiv.), 1,2-dimethylimidazole (0.01 equiv.), and phenylacetylene (-1.05 equiv.) for Cu-catalyzed azide-alkyne cycloaddition (CuAAC)[64].

The resulting solution was purified using HPLC with a ZORBAX SB-C18 Semi-Preparative 9.4 × 250 mm, 5 μm column, and freeze-dried for the LC-MS (ES-API) experiment, calculated for $C_{14}H_{17}N_3O_5^+$ [M + H]$^+$: $m/z$ = 308.12, observed: 308.1 (Supplementary Fig. 17b) and $^1$H NMR (Agilent) (Supplementary Fig. 17c). $^1$H NMR (CD$_3$OD, 400 MHz) δ 8.57 (s, 1H), 7.87 (m, 2H), 7.46 (m, 2H), 7.37 (m, 1H), 5.65 (d, 1H), 3.99–3.90 (m, 2H), 3.74 (dd, 1H), and 3.64–3.50 (m, 3H) (Supplementary Fig. 17c,

top). As a reference, 1-azido-1-deoxy-glucose (Synthose) was purchased and reacted with Cu(I) and phenylacetylene as described above to synthesize 1-β-D-glucopyranosyl-4-phenyl-1H-1,2,3-triazole (Supplementary Fig. 17c, bottom). The NMR data were acquired using VnmrJ software version 4.2 and analyzed using MestReNova x64-14.2.0.

## Docking simulation using Autodock Vina
We obtained a substrate coordinate file from the PubChem database as reported previously[65]. Then, we manually mutated the X-ray crystal structures of OmpF1 and OmpF2 to account for the mutations acquired from rational redesign and directed evolution using the PyMOL program. After setting these residues to be flexible, we applied the files to the Autodock Vina software version 1.1.2 for docking simulation[66] to simulate and visualize the potential binding poses of 4-β-MUG in the evolved OmpF variants.

## Reporting summary
Further information on research design is available in the Nature Portfolio Reporting Summary linked to this article.

## Data availability
All the data generated in this study are available within the main text, the Supplementary Information file; source data are provided in the Source Data file. Data are also available from the corresponding author upon request. Coordinates and structure factors for the reported crystal structures in this work were deposited in the RCSB PDB under accession numbers 7FDY (OmpF1) and 7FF7 (OmpF2). Source data are provided with this paper.

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

## Acknowledgements

This work was supported by the Creative-pioneering researchers program from Seoul National University (SNU to W.J.S.) and National Research Foundation from Korea government (NRF-2019R1C1C1003863 and NRF-2022R1A2C4001207 to W.J.S.).

## Author contributions

W.J.J. and W.J.S. designed the project, W.J.J. performed the experiments, and W.J.J. and W.J.S. wrote the paper.

## Competing interests

The authors declare no competing interests.
