## [Peer Review File · Nature Communications]

Design and Directed Evolution of Noncanonical β - Stereoselective MetalloglycosidasesReviewers' comments:

Reviewer #1 (Remarks to the Author):

This manuscript describes the construction of a Zn-binding site in OmpF protein and directed evolution to increase its hydrolytic(?) activity toward 4-methylumbelliferyl beta-D-glucoside. I do not judge if the catalytic proficiency of 2.8×10^9 is high enough for de novo designed enzymes. However, I would like to say that the non-enzymatic hydrolysis of glycosidic bonds is very slow and glycosidases can speed up the reaction by a factor of more than 10^{17} (Wolfenden et al., J. Am Chem. Soc. 120, 6814-6815, 1998). I have several comments on this paper.

Major points:

1. The quality of crystallographic analysis, especially for the resolution, is too bad to evaluate the zinc-binding site. The validation reports (global validation metrics) indicate that they are far worse than average entries determined at a similar resolution.
2. Figure 2. Types of electron density maps ($2m|Fo|-D|Fc|$, omit, or polder maps?) and their contour levels are not written. The quality of the figure (especially for the protein side chain) is too bad to understand if the OmpF1 protein really has a 3His coordination.
3. The crystallographic resolution is not enough to assign the fourth coordinating atom (water or chloride ion).
4. Extended Data Table 1. It must contain CC-half values. The average B-factors of the refined structures (7FDY and 7FF7) are too high. Usually, atoms with a B-factor higher than 80 \AA^2 should be omitted.
5. To support the hypothetical binding mode in Figure 16, a mutational analysis must be shown.
6. Figures 3a, 3b, and 3e must have WT as a negative control.
7. L87-88. "All variants exhibited Zn-dependent esterase activities". I could not find dotted lines (-Zn) in Figure 3a.
8. Figure 3. I could not find "P and /E", which is written in the legend.
9. Supplementary Table 4a contains both s-1 and min-1 as kinetic parameters. I prefer using s-1 for enzymes.
10. The TON value in Figure 4 has no unit (per second or minute?).
11. It is very unusual that the zinc-coordinated water acts as a catalytic acid (proton donor) in hydrolysis by metalloenzymes (Figure 4 and main text). I am not convinced by this proposed reaction mechanism. To test this hypothesis, I strongly suggest the authors do a pH-profile analysis of this hydrolysis.
12. Why did the authors use azide and click chemistry to assign the anomer retaining/inverting type (Supplementary Figure 13 and L157-161)? I think the presence of a high concentration of external nucleophile may change the "reaction mechanism" of this weak "enzyme".
13. Figure 4. If a retaining glycoside hydrolase is inhibited by the irreversible epoxide inhibitor like CBE, it is linked to the side chain of catalytic nucleophile (Glu or Asp) in many cases (PMID: 31419756). Figure 4f looks inconsistent with the proposed reaction mechanism in Figure 4d.
14. The order of the appearance of Supplementary Figures (e.g., Figures 1 and 2) is not correct. The manuscript seems to have been rewritten from a style of "Methods section before Results".
15. The activity assay buffer contains octyl-beta-glucoside (L314). I wonder if this protein also cleaves this detergent.

Reviewer #2 (Remarks to the Author):

This manuscript by Jeong and Song reports a novel artificial metalloenzyme with esterase, β -lactamase, and glycosidase activities. Being the last one, the first example of an artificial metal-dependent glycosidase developed in a protein space. This is potentially interesting due to the importance of developing novel glycosidases for the conversion of saccharides biomass or for biomedical purposes.

The rationale behind of the structural-guided design of the active site in the OmpF protein chosen as

scaffold appears to have been performed carefully, supported from previous examples of Zn-binding metalloproteins (supplementary table 1), and contains the appropriate supporting information which is well presented. Similarly, the re-design of OmpF variants to improve activity by incorporating a pair of acidic residues near the Zn-OH₂ seems clever based on mechanistic and geometric parameters (supplementary table 3). In terms of hydrolytic activities, all OmpF variants seems to exhibit similar or even higher Zn-dependent esterase and β -lactamase activities compared with featured examples in the literature. However, only moderate carbohydrate cleavage activity has been described compared with related peptide-based artificial glycosidases. Yet, this work builds upon a completely different and challenging protein-based scaffold, increasing the synthetic scope in the field of artificial metalloenzymes. Overall, publication in Nature Communication is recommended after addressing the several important concerns below:

Comment 1. I would recommended remove the expression “de novo” from the title and discussion. “De novo” design is related to the development of peptidic or protein systems that are not found in nature and fold into a stable and active scaffold. The novel designed Zn-binding active site constructed here is based on an already native protein domain (OmpF protein). I recommend that this work should be considered as a rational re-design of an existing protein scaffold with novel activities, and, in consequence, the discussion should be modified in this sense. The title seems to be too general.

Comment 2. Even if the X-Ray structure in Zn-bound state confirm a refolding in a β -barrel trimer construction, the SDS page of the Figures S3b and S7 display a 75kDa band, while to be a trimer the band should be closer of 110 KDa.

Comment 3. I would recommend a clearer draw of a figure showing the opposite orientation toward periplasmic side in the outer membrane in OmpF1 and 2.

Comment 4. ChemDraw structure of Fig. 3a is covering the diagram black line.

Comment 5. Legends in supplementary tables 4 and 5 should be provided. Also, same format to indicate confidence intervals should be provided on both cases. The number of replicates should be indicated for the designed experiments.

Comment 6. In the manuscript, the authors claim in the lines 92-95: “These data suggest that their hydrolytic activities are presumably determined by various factors, such as the first coordination spheres (3His versus 2His/1Glu), the directionality of Zn-bound water molecules, and surrounding microenvironments”. How the pH affects to the hydrolytic activities?

Comment 7. In the lines 99-102, the authors claim: “Because OmpF is the native passage of antibiotics and metabolites into the cells, it is possible that the intrinsic binding affinity of β -lactam to OmpF might have benefited the emergence of artificial β -lactamases with substantial catalytic activity without any sequence optimization.” Which is the binding affinity of nitrocefin to OmpF (KD)? And, how different is with respect to 4- β -MUG? this binding affinity is affected by the OmpF variants in the case of the glycosidase activity?

Comment 8. In the Supplementary Figure 8c should be indicated the correlation of each image with the sample described (upper and bottom).

Comment 9. In the lines 152-156: “When 4- α -MUG with an α -1,4-glycosidic linkage was employed instead, all evolved OmpF variants displayed nearly no time-dependent fluorescence increase. These data demonstrate that they are highly β -stereoselective glycosidases (Fig. 4b, Supplementary Fig. 11), presumably due to the intrinsically chiral nature of the protein environments and selection procedures only with β -glycosides as the screening substrates” Data for the binding affinity of the α -glycosidase compared to the β isomer (comment 7) are highly recommended to further support this claim.

Comment 10. In the lines 161- 164, the authors proposed that: “the role of the Zn site by revising the Koshland mechanism as follows (Fig. 4d); the newly designed Zn site replaces one of the canonical acid pairs. Then, Zn-OH₂/OH site may operate as a Lewis acid/base when an adjacently located acidic residue, possibly D113E, E117, or E62, acts as a nucleophile”. Which is the pKa of the Zn-H₂O site of the different evolved OmpF variants? A pH and pKa study should be presented to further explain the catalytic role of the Zn-OH₂/OH site operating as a Lewis acid/base pair (related to comment 6).

Comment 11. In the methods section “In vitro esterase and β -lactamase activity assays” the authors describe the use of 1.2 equiv. of Zn ion per monomer without specify which specie (ZnCl₂??). Same occur in the supplementary figure 5 and 6.

Comment 12. How the Zn loading in cells was determined? Have Zn affinities for the OmpF variants been determined? It would be interesting to compare this data with native Zn-binding proteins.

Reviewer #3 (Remarks to the Author):

In their article "De Novo Design and Evolution of Noncanonical Metalloglycosidases", the authors W.J. Jeong and W.J. Song artificially designed de novo metalloglycosidases by inserting a Zn-binding site in the membrane protein OmpF.

Since natural metalloglycosidases were not yet discovered, this work extends the access towards glycosidic bond hydrolysis via metallocofactors by enzymatic means as compared to synthetic inorganic complexes.

This is another great work of how experimental design & testing can produce novel, non-natural enzymatic reactivity from (in this case) a protein of previously different functionality.

The study is novel and well organised, the writing requires revision, but the work is presented in nice figures. Generally, the data should be sufficient proof that they succeeded in what they claim, particularly, because they determined crystal structures with zinc bound in their newly installed binding site of the de novo protein — I only have one concern expressed in 1) below. In addition, they added a great colour-based growth-dependent directed evolution approach to improve the activity that they had generated de novo.

1) There is only the aspect of apo-state activity:

The subheading in line 86 ("In vitro Zn-dependent hydrolytic activities of OmpF variants") already implies that here the Zn-dependency will be proven. However, in lines 86-88 it is only stated that "Zn-dependent activity is exhibited", not how this observation was tested. Were control experiments executed with buffer lacking zinc-ions?

In addition, in lines 128-130 the authors state that "Of note, OmpF1/E, but not others, exhibited fluorescence increase even as the apo-state, implying that alternative Zn-independent catalytic pairs might have co-emerged serendipitously along with the Zn-mediated ones" — this is an important statement, that is not discussed further within the manuscript, but should be as shown in my next paragraph:

Although the authors try to reason the Zn-dependency once again towards the end from line 170 to 183, the zinc-lacking ("-Zn") reactions shown in Figure 4e are of substantial significance: the TON of OmpF2/E-R2 (green) is only twice as high with ("+Zn") than without Zn, and ca. 3-fold for variant OmpF1/E-R4 (black). This raises questions and concerns, and should be addressed by the authors in the manuscript. But these observations are barely mentioned, it is only stated in lines 166-167 that "The evolved OmpF variants exhibited significantly higher TON in the presence of Zn than in the apo-state".

2) While reading the manuscript, I wondered about the overall impact of this study: We know from literature (including work from the author's research group) that the de novo design of such artificial enzymes is possible. Clearly, more and more examples of new reactivity or artificial enzymes will be thrilling to see.

But are such examples necessary? Will the work be of significance to the field? Why is this particular metalloglycosidase in this study important? What is the use of it? Unprecedented perhaps, but also useful? How do they compare? This message is not yet clear to me.

Therefore, I was slightly irritated to only find a "discussion" of less than 10 lines in size! It does not include any reference and sounds like a quickly written conclusion.

Although I know that a discussion part for Nature Communications is not strictly necessary, a succinct one is included in most of its published work. I would highly recommend to do the same in this manuscript ("succinct" does not exclude "significant" and "comprehensive").

3) Will this artificial enzymes be chemically useful? What (extended) substrate scope do the variants have? Are the reported catalytic proficiencies superior to already existing tools?

While I appreciate the up-scaling effort via click chemistry plus NMR characterisation (infos missing, see 4) I) below), the authors almost exclusively used highly reactive phenol-glucoside test substrates, which possess extremely good leaving groups. This feigns high catalytic activity — how does the reactivity towards "real" alkylglycosides (e.g. methyl glucoside) compare? I assume efficiencies will be much smaller...

Furthermore, they only compare the catalytic efficiency of the variants to their own published work of artificial beta-lactamases (line 98). In fact, the "catalytic proficiency of up to 2.8×10^9 " in the abstract only gets meaning when reading line 150 relating this value as an enhancement from the uncatalysed rate of glycosidase activity.

The following references or others alike may be of use for this:

- Cowan J. A. et al., 2021, 10.1021/acs.inorgchem.0c01193
- Cowan J. A. et al., 2020, 10.1002/smll.202000392
- Nelson A.-G. D. et al., 2010, 10.1021/ic9014064
- Cowan J. A. et al., 2017, 10.1002/anie.201612079

I believe such information and wider perspective is highly necessary to be communicated to the scientific community in order for this manuscript to be of the high impact that Nature Communication embodies.

4) Furthermore, I highly recommend the authors to check the manuscript again and allow an English native speaker proofread the manuscript. There are many grammar / language mistakes or hyphenation inconsistencies, for example:

a) whole cell & whole-cell (throughout the manuscript, e.g. page 3, introduction, line 47 and Figure 3 caption, line 595)
should be "whole-cell" (when used as adjective)

b) present/ past tenses throughout the text, for example in lines 45-46
"This discrepancy leads us to question whether metal-dependent glycosidases can be synthesized"
should be "led"

c) "All variants exhibited Zn-dependent esterase activities, demonstrating that catalytically active Zn-binding motifs are created within the outer membrane protein"
should be "were created"

d) "The kinetic parameters of OmpF variants are similar, while their catalytic efficiency and substrate-binding affinity were substantially higher than those of artificial β -lactamases with analogous Zn sites."
should be "were similar" just as "were substantially higher"

e) line 177
"supporting that the Zn site show sufficient basicity"
should be "showed" (or "shows" if present tense was desired)

f) Repetitions do not read well
lines 57, 62 & 64: "Next", "Then", "Then"

g) line 66-67:
"Both variants were obtained by E. coli heterologous expression"
should be "heterologous expression in E. coli"

h) lines 192-194
"This work also provides a novel molecular basis to design and develop versatile biocatalysts to reengineer metabolic pathways and to produce biofuels with glycosides."
should be "biocatalysts" (comma missing after "biocatalysts")

i) Article missing

"To bypass refolding step, we herein isolated"
should be "To bypass the refolding step"

j) line 150

"uncatalysed rate of 4-methylumbelliferyl- β -D-glucopyranoside"
a compound has no rate, do you mean the glycosidation?^[1]_[SEP]

k) line 113-116

"To improve their basal activities of OmpF variants, we integrated the canonical Koshland mechanism that inverting and retaining β -glycosidases operate via a pair of acidic residues 6–11 Å and 5.1–5.5 apart, respectively"

This sentence needs revising.

l) line 410

Did you measure the ¹H NMR at 400 MHz or 500 MHz? Both values are given. What product mass did you inject and what yield does it equal to?

m) Figures

Usually, the figure captions come below the figure (in contrast to tables and their captions)

In the following paragraphs, we address the reviewers' comments and detail specific changes made to the manuscript.

Referee 1

This manuscript describes the construction of a Zn-binding site in OmpF protein and directed evolution to increase its hydrolytic(?) activity toward 4-methylumbelliferyl beta-D-glucoside. I do not judge if the catalytic proficiency of 2.8×10^9 is high enough for de novo designed enzymes. However, I would like to say that the non-enzymatic hydrolysis of glycosidic bonds is very slow and glycosidases can speed up the reaction by a factor of more than 10^{17} (Wolfenden et al., J. Am Chem. Soc. 120, 6814-6815, 1998). I have several comments on this paper.

We agree with the reviewer's comment that the high catalytic proficiency of the OmpF variant is not solely owing to its high reactivity of the enzymes. It is the consequence of its enhanced reactivity from iterative sequence optimization and slow uncatalyzed reaction rate. Because a slow catalytic rate indicates that the glycosidic bond cannot hydrolyze, the high catalytic proficiency suggests that our work successfully tackled the reaction and expanded the chemical scope of artificial metalloenzymes.

We agree with the reviewer's comment that our artificial metalloenzyme is far less reactive than natural enzymes. However, artificial enzymes that we report herein are born from only two to four rounds of screening and selection. In contrast, natural enzymes have evolved for millions of years. What we intend to do here is not a competition against natural enzymes. Instead, we aimed to demonstrate that current-state-of the art in metalloenzyme design can lead to the birth of metal-dependent glycosidases that function via a non-canonical mechanism and reveal the emergence of unprecedented inorganic reactivity in protein environments. Therefore, we believe that our work has extraordinary value, originality, and potential for publication.

Q1) The quality of crystallographic analysis, especially for the resolution, is too bad to evaluate the zinc-binding site. The validation reports (global validation metrics) indicate that they are far worse than average entries determined at a similar resolution.

We appreciate the reviewer's comment. We refined the X-ray crystal structures of both OmpF variants to improve the quality of the datasets. Due to the relatively low resolution of the datasets, the global validation metrics seem only minimally improved. However, newly refined data now have reasonable B-factors. We also included CC-half values, 100% (96) for OmpF1 and 98% (98) for OmpF2, in the revised Extended Data Table 1.

In addition, the electron density maps overlaid with anomalous difference maps shown in Fig. 2b–c clearly demonstrate that OmpF variants possess coordinatively unsaturated Zn-binding sites.

Q2) Figure 2. Types of electron density maps ($2m|Fo|-D|Fc|$, omit, or polder maps?) and their contour levels are not written. The quality of the figure (especially for the protein side chain) is too bad to understand if the OmpF1 protein really has a 3His coordination. In the revised manuscript, we included the description for the electron density maps to read, “The gray and blue grid represents the $2Fo - Fc$ electron density contoured at 1.0σ and anomalous difference maps contoured at 5.0σ in OmpF1 and 3.0σ in OmpF2, respectively.”

We also created an additional figure to visualize a tetrahedral Zn-site composed of 2His/1Glu motif in OmpF1, as shown below (left). The electron density of non-ligating H102 and metal-ligating E71'/H82/H132 residues is colored in cyan and magenta, respectively. Although the resolution of the datasets is relatively low, there is no detectible electron density between H102 and Zn ion. Consequently, the first coordination sphere of OmpF1 is noticeably different from that of OmpF2 (right), where 3His (H83/H102/H132) residues are ligated to the Zn ion.

Q3) The crystallographic resolution is not enough to assign the fourth coordinating atom (water or chloride ion).

We agree with the reviewer’s comment that the resolutions of our X-ray crystal structures are relatively low and it is not feasible to distinguish water from chloride ion. Therefore, we removed a chloride ion and refined both datasets as addressed above in **Q1**. The newly refined datasets demonstrate that two out of three protomers in OmpF1 and

OmpF2 have additional electron density at the Zn sites, which can be now fitted with water molecules. We also revised the paragraph to describe the X-ray crystal structures in the main text to read, “...**Zn-binding site. Regardless, the fourth site was ligated by a non-proteinaceous molecule, tentatively assigned as a water or unoccupied.**” and “...**The fourth coordination site was coordinatively unsaturated, and therefore, OmpF2 is also suitable for our studies.**”

Q4) Extended Data Table 1. It must contain CC-half values. The average B-factors of the refined structures (7FDY and 7FF7) are too high. Usually, atoms with a B-factor higher than 80 Å² should be omitted.

As recommended by the reviewer, we further refined the structures of OmpF variants, and we omitted the atoms with unreasonably high B-factor values. Although refinements significantly improved the statistics (For example, the average B-factors for water molecules are 65.22 and 62.44 in OmpF1 and OmpF2 structures, respectively), the overall B-factors are still relatively high. It is presumably due to the low resolution of the original datasets. However, at least one of the protomers has modest data qualities and statistics; they show the B-factors of 96.92 and 83.56 in OmpF1 and OmpF2, respectively. We are currently under further investigation to obtain better datasets by optimizing protein purification and crystallization procedures.

Q5) To support the hypothetical binding mode in Figure 16, a mutational analysis must be shown.

We agree with the reviewer that mutation studies are necessary to propose the residues that might form substrate-binding sites. OmpF1/E-R4, OmpF1Y/E-R4, and OmpF2/E-R2 variants are derived from the iterative saturated mutagenesis and screening. Therefore, the experiments can be taken as an indirect way to validate the mutation effects. In addition, other acidic residues nearby the Zn-sites were mutated into aspartate or alanine (E113D, D113A, and E117A in OmpF1 and E113A in OmpF1Y), and the reduction or disappearance of catalytic activities was reported in Figure S13. In addition, we prepared three variants of OmpF2 (D113A, E117A, and E62A) and measured the catalytic activity. These data suggest that these residues can be strongly involved in determining the Zn-dependent catalytic activities. They are consistent with our previous description, the docking data shown in Figure S18, and two proposed mechanisms shown in Figure 5C in the revised manuscript.

In the revised manuscript, we revised the paragraphs that address the mutational studies to read,

“...In addition, D113E conservative mutation (denoted as /E) modulated the glycosidase activities of OmpF variants; OmpF1/E and OmpF2/E showed substantially elevated activities...”

“The removal of Zn ion or the mutation of adjacently located acidic residues, such as E113, E117, and E62 (Supplementary Fig. 13), partially or completely inactivated the enzymes, suggesting that a Zn-site and at least one acidic residue in proximity constitute a noncanonical catalytic motif of Zn-dependent glycosidases.”

Q6) Figures 3a, 3b, and 3e must have WT as a negative control.

In the initial version of the manuscript, the data for the wild-type protein were not shown only because they are inactive or negligible in all activity assays. The revised manuscript addressed no detectable background activity of the wild-type protein as follows, “...**The wild-type protein shows no detectable Zn-dependent hydrolytic activities. In contrast, all Zn-complexed OmpF variants...**”

Q7) L87-88. “All variants exhibited Zn-dependent esterase activities”. I could not find dotted lines (-Zn) in Figure 3a.

We apologize for the confusion. The figure is now revised to read, “**Solid and dotted lines in c indicate the presence and absence of Zn ions, respectively.**”

The Zn-free background hydrolytic activities are included in Supplementary Figure 5, and only the net activities are included for clarity.

Q8) Figure 3. I could not find “P and /E”, which is written in the legend.

We initially included the description of “P and /E” in the figure 3 legend as “where P and /E indicate the parent proteins and D113E mutation, respectively.”

In the revised manuscript, we revised the figure legend as “**4-β-MUG, where P and /E indicate the parent proteins (OmpF1, OmpF1Y, and OmpF2) and their D113E single-mutants, respectively.**”

We also edited the main text to read, “...**They showed significantly elevated whole-cell activities with the initial rates up to two orders of magnitude relative to those of their rationally designed parent proteins (P; OmpF1, OmpF1Y, and OmpF2, respectively).**”

Q9) Supplementary Table 4a contains both s^{-1} and min^{-1} as kinetic parameters. I prefer using s^{-1} for enzymes.

According to the reviewer’s comment, we converted all kinetic parameters, as shown below.

(a)

	k_{cat} (s^{-1})	K_{M} (mM)	$k_{\text{cat}}/K_{\text{M}}$ ($\text{s}^{-1} \text{M}^{-1}$)
OmpF1	0.006 ± 0.001	0.5 ± 0.4	12 ± 8
OmpF1Y	0.026 ± 0.004	2.5 ± 0.8	10 ± 2
OmpF2	0.030 ± 0.004	3 ± 1	12 ± 3

(b)

	k_{cat} (s^{-1})	K_{M} (mM)	$k_{\text{cat}}/K_{\text{M}}$ ($\text{s}^{-1} \text{M}^{-1}$)
OmpF1	0.017 ± 0.004	1.4 ± 0.6	12 ± 2
OmpF1Y	0.02 ± 0.01	2 ± 1	15 ± 3
OmpF2	0.025 ± 0.005	1.7 ± 0.6	14 ± 2

Q10) The TON value in Figure 4 has no unit (per second or minute?).

The TON value herein indicates the number of moles of products per the number of moles of enzymes, not a kinetic constant (k_{cat}) from steady-state activity assays. Therefore, the

TON value has no unit. For clarity, we revised the main text to read, “...We also measured turnover number (TON) of the evolved variants with 4- β -MUG, the mole of products per that of enzyme, using cell lysates (Fig. 4c).”

Q11) It is very unusual that the zinc-coordinated water acts as a catalytic acid (proton donor) in hydrolysis by metalloenzymes (Figure 4 and main text). I am not convinced by this proposed reaction mechanism. To test this hypothesis, I strongly suggest the authors do a pH-profile analysis of this hydrolysis.

We agree with the reviewer’s comment that Zn-coordinated water usually functions as a nucleophile rather than Lewis acid. However, if Zn-OH functions as a nucleophile, the cleavage of the C–O bond between the anomeric carbon of glucose and the Zn-bound oxygen atom is required, as described in Figure 5c (right) in the revised manuscript. Although further investigation is required, we thought that it seems to be less likely to occur.

We agree with the reviewer’s suggestion that pH-dependent experiments can be informative in proposing a reaction mechanism. Natural glycosidases often show a bell-shaped pH-dependent activity, resulting in the optimal condition of pH 5–6.5 and two pK_a values accounting for a nucleophile and a Lewis acid/base.

Therefore, we conducted the pH-dependent activity assays of all three evolved variants (OmpF1/E-R4, OmpF1Y/E-R4, and OmpF2/E-R2) in pH 6.0–9.0. With the Zn-complexed variants, bell-shaped curves were observed. Significantly elevated optimal pH conditions with two pK_a values ($pK_{a1} = 7.7–7.9$ and $pK_{a2} = 8.2–8.3$) were detected. In addition, the apo-proteins show negligible or less noticeable pH dependence, implicating that at least one of the pK_a values detected from the Zn-bound proteins is likely to be associated with the Zn-site.

Both pK_a values can be derived from Zn-OH₂/OH, which is comparable to those in artificial metalloproteins ($pK_a = 8.4–9.2$) and metallo-peptides ($pK_a = 8.2–9.6$), and the local chemical environments of OmpF may alter the chemical properties significantly. Therefore, these results may not differentiate the role of the Zn-site between a nucleophile and a Lewis acid/base.

Regardless, these results indicate the unprecedented metalloglycosidases are mediated by two catalytically essential motifs analogous to the natural enzymes. Therefore, we included the pH-dependent catalytic activities of OmpF variants in figure 4d and Supplementary Fig. 15 in the revised manuscript.

We also inserted the following section in the revised manuscript to read,
“All three evolved OmpF variants showed pH-dependent activities with 4- β -MUG (Fig. 4d and Supplementary Fig. 15). Both Zn-complexed and apo-states show bell-shaped pH-dependence, revealing at least two ionizable side-chains to be essential for catalysis ($pK_a = \sim 7.7$ and 8.3). It is consistent with natural glycosidases of having two discrete pK_a values^{29,30}. However, Zn-complexed and the apo-state of OmpF variants yielded the maximal TONs at pH 8.0–8.5 and 7.5–8.0, respectively, whereas natural glycosidases show the pH optimum at pH 5.0–6.5^{30,31}. The significant alteration in pK_a values might be derived from replacing one of an acidic pair with a Zn-OH₂/OH moiety. Besides, the unique chemical environments of OmpF might account for the pH-dependence; it was reported that the pK_a values of the side-chains in the constriction zone are considerably perturbed from those of amino acids in bulk solvents³²⁻³⁴.”

We also modified the proposed mechanism, shown in Figure 5c in the revised manuscript. We listed both potential mechanisms, where the Zn-site play as a Lewis acid/base or Lewis/nucleophile (left) or a nucleophile (right).

In addition, we included the paragraph that describes the proposed chemical mechanism of OmpF variants as follows, **“Therefore, we proposed a reaction mechanism of OmpF2/E-R2 by revising the Koshland mechanism in that one of the canonical acid pairs is replaced by a coordinatively unsaturated Zn-site (Fig. 5c). It is possible that an acidic residue, possibly D113E, initiates glycosylation as a nucleophile when Zn-OH₂ species acts as a Lewis acid. Then, the resulting Zn-OH may become a direct nucleophile or activate a water molecule as a Lewis base, facilitating deglycosylation step. Alternatively, the role of an acidic residue and Zn-OH₂ moiety might be the opposite of each other, although, in this case, the scission of C–O bond in deglycosylation is necessary to release β -glycosides from the Zn-site.”**

Q12) Why did the authors use azide and click chemistry to assign the anomer retaining/inverting type (Supplementary Figure 13 and L157-161)? I think the presence of a high concentration of external nucleophile may change the “reaction mechanism” of this weak “enzyme”.

We agree with the reviewer’s comment that excess amounts of azide may alter the relative orientations of the residues and the bound substrates, thus changing the reaction mechanism. However, the azide reaction is a well-established and commonly conducted assay to identify the chemical mechanism of natural glycosidases. To provide a more biochemical basis for this experiment, we included three references and revised the manuscript to read,

“We also analyzed the structure of hydrolyzed glycoside in the presence of excess azide, which functions as an external nucleophile for mechanistic studies, as described previously^{30,37,38} (Supplementary Fig. 17). OmpF2/E-R2 yielded only glucose, which is not suitable for structural analysis. In contrast, OmpF2/E-R2* variant (Y102H/R82Y), which differs from OmpF2/E-R2 only by a single residue at the position 82, produced the mixtures of glucose and 1-azido-1-deoxy-glucose, the latter of which can be isolated.”

Q13) Figure 4. If a retaining glycoside hydrolase is inhibited by the irreversible epoxide inhibitor like CBE, it is linked to the side chain of catalytic nucleophile (Glu or Asp) in many cases (PMID: 31419756). Figure 4f looks inconsistent with the proposed reaction mechanism in Figure 4d.

We apologize for the confusion. Figure 4f is only one of the several fragmentations we have obtained, and the rest are listed in Supplementary Table S7. For clarity, we included additional data into Fig. 5 in the revised manuscript, as shown below.

Q14) The order of the appearance of Supplementary Figures (e.g., Figures 1 and 2) is not correct. The manuscript seems to have been rewritten from a style of “Methods section before Results”.

We revised the manuscript accordingly.

Q15) The activity assay buffer contains octyl-beta-glucoside (L314). I wonder if this protein also cleaves this detergent.

It is an exciting idea, and we appreciate the reviewer’s comment. As suggested by the reviewer, some natural glycosidases show catalytic activities with octyl-β-glucoside. Therefore, we experimented to identify whether octyl-beta-glucoside can be a substrate

with OmpF2/E-R2. We observed the formation of glucose and 1-octanol by LC and GC, respectively, as included in Supplementary Fig. 14.

Therefore, we included the results in the revised manuscript to read, **“In addition, we observed that OmpF2/E-R2 variant is capable of hydrolyzing the β -glycosidic bond in n-octyl- β -D-glucopyranoside (OG), which was initially added as a nonionic detergent for the preparation of membrane proteins. Formation of glucose and n-octanol were detected as the hydrolyzed products (Supplementary Fig. 14), indicating that the evolved OmpF variant accommodates a hydrophobic and bulky substrate that was not even used for selection.”**

Reviewer #2:

This manuscript by Jeong and Song reports a novel artificial metalloenzyme with esterase, β -lactamase, and glycosidase activities. Being the last one, **the first example** of an artificial metal-dependent glycosidase developed in a protein space. This is **potentially interesting** due to the importance of developing novel glycosidases for the conversion of saccharides biomass or for biomedical purposes. The rationale behind of the structural-guided design of the active site in the OmpF protein chosen as scaffold appears to have been performed carefully, supported from previous examples of Zn-binding metalloproteins (supplementary table 1), and contains the appropriate supporting information which is **well presented**. Similarly, the re-design of OmpF variants to improve activity by incorporating a pair of acidic residues near the Zn-OH₂ seems **clever** based on mechanistic and geometric parameters (supplementary table 3). In terms of hydrolytic activities, all OmpF variants seems to exhibit similar or even higher Zn-dependent esterase and β -lactamase activities compared with featured examples in the literature. However, only moderate carbohydrate cleavage activity has been described compared with related peptide-based artificial glycosidases. **Yet, this work builds upon a completely different and challenging protein-based scaffold, increasing the synthetic scope in the field of artificial metalloenzymes. Overall, publication in Nature Communication is recommended after addressing the several important concerns below:**

We appreciate the reviewer's comment that we have expanded the scope of artificial metalloenzymes. All concerns are addressed as shown below.

Q1) I would recommended remove the expression "de novo" from the title and discussion. "De novo" design is related to the development of peptidic or protein systems that are not found in nature and fold into a stable and active scaffold. The novel designed Zn-binding active site constructed here is based on an already native protein domain (OmpF protein). I recommend that this work should be considered as a rational re-design of an existing protein scaffold with novel activities, and, in consequence, the discussion should be modified in this sense. The title seems to be too general.

We appreciate the reviewer's comment. We used the term "de novo" to highlight that a "metal-ligating active site" is created to a non-related protein scaffold, different from repurposing a pre-existing metalloenzyme. Regardless, we agree with the reviewer's comment that the expression might be confusing. Therefore, we changed the title of our manuscript to "**Design and Directed Evolution of Noncanonical β -Stereoselective Metalloglycosidases.**"

Q2) Even if the X-Ray structure in Zn-bound state confirm a refolding in a β -barrel trimer construction, the SDS page of the Figures S3b and S7 display a 75kDa band, while to be a trimer the band should be closer of 110 KDa.

We apologize for not providing sufficient information on the biochemical properties of OmpF in the original manuscript. It was previously reported that OmpF oligomers are stable even in SDS detergents, and the properly refolded OmpF moves faster than the theoretical size of the trimeric state. If OmpF is not folded, it will run at a size of a monomer in SDS-PAGE. To avoid any confusion, we included the following sentence and reference in the figure legend to read,

“OmpF refolding from an unfolded monomer (left) to a folded trimer (right). As reported previously in reference 18, a properly-refolded OmpF trimer runs faster (80 kDa) than its molecular size (110 kDa) in SDS-gel.”

Q3) I would recommend a clearer draw of a figure showing the opposite orientation toward periplasmic side in the outer membrane in OmpF1 and 2.
We modified the figure legends in Fig. 2a and included additional figure (Figure S4A) in the revised manuscript to show both orientations.

Q4) ChemDraw structure of Fig. 3a is covering the diagram black line.
As suggested, we modified the Fig. 3a in the revised manuscript.

Q5) Legends in supplementary tables 4 and 5 should be provided. Also, same format to indicate confidence intervals should be provided on both cases. The number of replicates should be indicated for the designed experiments.
We added the legends in the Supplementary tables 4 and 5 in the revised manuscript. The number of replicates is included as well.

Q6) In the manuscript, the authors claim in the lines 92-95: “These data suggest that their hydrolytic activities are presumably determined by various factors, such as the first coordination spheres (3His versus 2His/1Glu), the directionality of Zn-bound water molecules, and surrounding microenvironments”. How the pH affects to the hydrolytic activities?

As mentioned by the reviewer, metal-mediated hydrolytic activities are pH-dependent because it alters the relative concentrations of M-OH₂ versus M-OH species. The revised manuscript includes the pH-dependent catalytic activities of OmpF variants in Figure 4d and Supplementary Fig. 15, as described as the answer to **#Reviewer1-Q11**.

Q7) In the lines 99-102, the authors claim: “Because OmpF is the native passage of antibiotics and metabolites into the cells, it is possible that the intrinsic binding affinity of β -lactam to OmpF might have benefited the emergence of artificial β -lactamases with substantial catalytic activity without any sequence optimization.” Which is the binding affinity of nitrocefin to OmpF (KD)? And, how different is with respect to 4- β -MUG? this binding affinity is affected by the OmpF variants in the case of the glycosidase activity?
We appreciate the reviewer’s comments on the binding affinity of the substrates. To the best of our knowledge, there is no precedent measurement of the binding affinity of beta-lactams to OmpF. We speculated on intrinsic binding affinity because there are X-ray crystal structures of OmpF complexed with ampicillin (PDB code 4GCP) and carbenicillin (PDB code 4GCQ). We can estimate the binding affinity from our steady-state kinetic results, where a saturation behavior was detected as a function of substrate (nitrocefin) concentrations. The Michaelis-constant (K_M), related to the dissociation constant (K_D) was measured to be ~1 mM.

For clarity, we modified the sentence to read,

“Their catalytic efficiency or substrate-binding affinity were comparable or higher than those of Zn-complexes^{23,24} and artificial Zn-dependent β -lactamases,²⁰

although OmpF sequence was yet to be optimized. Notably, all OmpF show saturation curves, suggesting OmpF may have benefited to show at least modest binding affinity with β -lactams as the native passage of antibiotics into the cells²⁵. Then, these results implicate that OmpF is a versatile scaffold that functions as a host-like macromolecule and interacts with guest-like small molecules for catalysis.”

On the other hand, the kinetic analysis of 4- β -MUG showed no saturation behavior with all three evolved variants, suggesting that the substrate-binding event was not detected from the steady-state kinetic analysis.

Q8) In the Supplementary Figure 8c should be indicated the correlation of each image with the sample described (upper and bottom).

We appreciate the reviewer’s comments. We corrected the figure and labels in the revised manuscript.

Q9) In the lines 152-156: “When 4- α -MUG with an α -1,4-glycosidic linkage was employed instead, all evolved OmpF variants displayed nearly no time-dependent fluorescence increase. These data demonstrate that they are highly β -stereoselective glycosidases (Fig. 4b, Supplementary Fig. 11), presumably due to the intrinsically chiral nature of the protein environments and selection procedures only with β -glycosides as the screening substrates” Data for the binding affinity of the α -glycosidase compared to the β isomer (comment 7) are highly recommended to further support this claim.

We agree with the reviewer’s comment that binding affinity could be one of the primary factors determining stereoselectivity. In addition, the relative orientation of the substrates in the active site pocket could also be crucial whether they are placed in catalytically-relevant positions. We conducted a competition experiment to distinguish these possibilities by adding the mixtures of 4- α -MUG and 4- β -MUG to OmpF2/E-R2. The reaction rates were indistinguishable from those with 4- β -MUG alone, suggesting that OmpF2/E-R2 shows much weaker binding affinity with 4- α -MUG than 4- β -MUG.

Q10) In the lines 161-164, the authors proposed that: “the role of the Zn site by revising the Koshland mechanism as follows (Fig. 4d); the newly designed Zn site replaces one of the canonical acid pairs. Then, Zn-OH₂/OH site may operate as a Lewis acid/base when an adjacently located acidic residue, possibly D113E, E117, or E62, acts as a nucleophile”. Which is the pK_a of the Zn-H₂O site of the different evolved OmpF variants? A pH and pK_a study should be presented to further explain the catalytic role of the Zn-OH₂/OH site operating as a Lewis acid/base pair (related to comment 6).

We agree with the reviewer’s comment and we measured the pH-dependent catalytic activities as discussed above in **#Reviewer1-Q11**.

Q11) In the methods section “In vitro esterase and β -lactamase activity assays” the authors describe the use of 1.2 equiv. of Zn ion per monomer without specify which specie (ZnCl₂??). Same occur in the supplementary figure 5 and 6.

We appreciate the reviewer’s comment. We used zinc chloride (ZnCl₂), and we revised the manuscript to clarify the source of zinc ions to read, “**ZnCl₂ solution was added to**

the resulting protein at the ratio of 1.2 equiv. of Zn ion to the protomer.”, “a final concentration of 10 μM in 50 mM Tris-HCl (pH 8) buffer were pre-mixed with 1.2 equiv. of Zn ions (ZnCl_2) to OmpF monomer on a 96-well plate.”

Q12) How the Zn loading in cells was determined? Have Zn affinities for the OmpF variants been determined? It would be interesting to compare this data with native Zn-binding proteins.

We appreciate the reviewer's comment on Zn affinity. We conducted the ICP-MS analysis of OmpF2/E-R2 and the wild-type protein after directly extracting from the whole-cells. OmpF2/E-R2 is isolated with a substantial increase in Zn concentrations relative to the wild-type protein, at the molar ratio of approximately 1:1 Zn ion to protomer, suggesting that every Zn-binding site forms holo-states. Given that 50 μM ZnCl_2 -enriched growth media was used, it is likely that OmpF2/E-R2 exhibits sub-micromolar binding affinity. The results are analogous to the previous studies of artificial Zn-binding proteins, where the binding affinity was measured to be nM to μM (J. Am. Chem. Soc. 2010, 132, 25, 8610–8617; J. Am. Chem. Soc. 2013, 135, 15, 5895–5903). These data are included in the revised manuscript to read, “...**However, the inductively coupled plasma-mass spectrometry (ICP-MS) analysis suggests that OmpF2/E-R2 protein is embedded as in Zn-bound state on the outer membrane of *E. coli* cells (Supplementary Table 6), indicating that a coordinatively unsaturated Zn-site is the key player in whole-cell catalysis.**”

Reviewer #3:

In their article "De Novo Design and Evolution of Noncanonical Metalloglycosidases", the authors W.J. Jeong and W.J. Song artificially designed de novo metalloglycosidases by inserting a Zn-binding site in the membrane protein OmpF. Since natural metalloglycosidases were not yet discovered, this work extends the access towards glycosidic bond hydrolysis via metallocofactors by enzymatic means as compared to synthetic inorganic complexes.

This is another great work of how experimental design & testing can produce novel, non-natural enzymatic reactivity from (in this case) a protein of previously different functionality. The study is novel and well organised, the writing requires revision, but the **work is presented in nice figures**. Generally, the data should be **sufficient proof** that they succeeded in what they claim, particularly, because they determined crystal structures with zinc bound in their newly installed binding site of the de novo protein - I only have one concern expressed in 1) below. In addition, they added a **great colour-based growth-dependent directed evolution approach** to improve the activity that they had generated de novo.

We appreciate the reviewer's comment on the novelty, organization, sufficient experimental results, and screening system. All concerns are addressed as shown below.

Q1-1) There is only the aspect of apo-state activity: The subheading in line 86 ("In vitro Zn-dependent hydrolytic activities of OmpF variants") already implies that here the Zn-dependency will be proven. However, in lines 86-88 it is only stated that "Zn-dependent activity is exhibited", not how this observation was tested. Were control experiments executed with buffer lacking zinc-ions?

Initially, we included the data for both apo- and Zn-complexed proteins in the Supplementary, and only the net activities were shown as Zn-dependent activity in the main text. In the revised manuscript, we included how the *in vitro* Zn-dependent hydrolytic activities were determined to read,

"To determine whether OmpF variants generate a Zn-mediated nucleophilic site for hydrolysis, we first measured esterase activities with a chromogenic substrate, *p*-nitrophenyl acetate (pNPA), in the presence and absence of Zn ions, and obtained the difference as the Zn-dependent net activities (Fig. 3a, Supplementary Fig. 5, and Supplementary Table 4a)."

"...In contrast, all Zn-complexed OmpF variants exhibited considerably higher esterase activities than the apo-forms, demonstrating that hydrolytically active Zn-binding sites are created, similar to synthetic, peptide-, and protein-based catalysts¹⁹⁻²¹ and metalloesterases^{2,22}.

Q1-2) In addition, in lines 128-130 the authors state that "Of note, OmpF1/E, but not others, exhibited fluorescence increase even as the apo-state, implying that alternative Zn-independent catalytic pairs might have co-emerged serendipitously along with the Zn-mediated ones" - this is an important statement, that is not discussed further within the manuscript, but should be as shown in my next paragraph:

Although the authors try to reason the Zn-dependency once again towards the end from line 170 to 183, the zinc-lacking ("-Zn") reactions shown in Figure 4e are of substantial significance: the TON of OmpF2/E-R2 (green) is only twice as high with ("+Zn") than

without Zn, and ca. 3-fold for variant OmpF1/E-R4 (black). This raises questions and concerns, and should be addressed by the authors in the manuscript. But these observations are barely mentioned, it is only stated in lines 166-167 that "The evolved OmpF variants exhibited significantly higher TON in the presence of Zn than in the apo-state".

We appreciate the reviewer's comment on the catalytic activities of Zn-independent glycosidases. The revised manuscript includes the pH-dependent reactivities and LC-MS data of both Zn-bound and apo forms in Fig. 4d and Supplementary Figs. 15 and 16. In addition, we further discuss the co-emergence of Zn-dependent and Zn-independent glycosidases as follows,

"The iterative mutations also elevated the glycosidase activities of their apo-forms (Supplementary Table 5), suggesting that the alternative Zn-independent reaction routes have developed simultaneously via sequence optimization."

"However, Zn-complexed and the apo-state of OmpF variants yielded the maximal TONs at pH 8.0–8.5 and 7.5–8.0, respectively, whereas natural glycosidases show the pH optimum at pH 5.0–6.5^{30,31}. The significant alteration in pK_a values might be derived from replacing one of an acidic pair with a Zn-OH₂/OH moiety."

"In the apo-states, CBE was conjugated to D113E (Fig. 5b) or additional positions, such as D92 or E117. Because the wild-type protein and D113E single mutant show no such conjugation in the constriction zone, these data reveal that the evolved OmpF variants develop Zn-mediated nucleophilic sites that directly participate in the hydrolysis of glycosides."

Q2) While reading the manuscript, I wondered about the overall impact of this study: We know from literature (including work from the author's research group) that the de novo design of such artificial enzymes is possible. Clearly, more and more examples of new reactivity or artificial enzymes will be thrilling to see. But are such examples necessary? Will the work be of significance to the field? Why is this particular metalloglycosidase in this study important? What is the use of it? Unprecedented perhaps, but also useful? How do they compare? This message is not yet clear to me. Therefore, I was slightly irritated to only find a "discussion" of less than 10 lines in size! It does not include any reference and sounds like a quickly written conclusion. Although I know that a discussion part for Nature Communications is not strictly necessary, a succinct one is included in most of its published work. I would highly recommend to do the same in this manuscript ("succinct" does not exclude "significant" and "comprehensive").

Metalloenzyme catalysis has been primarily investigated by the discovery and characterization of natural enzymes and the synthesis of biomimetic complexes that resemble the structure or function of the active sites of natural enzymes. The second approach has been valuable because it enables conditions incompatible with proteins (low temperature, organic solvents, etc.), and the chemical space is nearly unlimited. However, the second approach could be limited to the synthesis of protein-like secondary coordination spheres. Along this line, we believe that the design and creation of artificial metalloenzymes can be an alternative approach to exploring how metalloenzyme works. The retrosynthesis and directed evolution of metalloenzymes allow us to validate how the chemical interplay between inorganic sites and surrounding protein environments dictates the overall structure and function of metalloenzymes.

In addition, we herein explored a fundamental question, “Why is there no such metalloglycosidase in Nature?” It is a genuinely essential question in bioinorganic chemistry because it can elucidate the unidentified chemical role and power of inorganic cofactors in proteinaceous environments and provide a reference to discover unexplored metalloproteomes. Given that we have created artificial metalloglycosidases, the absence of such an enzyme in Nature also becomes an intriguing topic for further investigation. It may provide information on the chemical environments (local pH conditions or metal abundance) where the canonical glycosidase has evolved.

Therefore, we rewrote the entire section of discussion as follows,

“The structure- and mechanism-based design of artificial metalloenzymes allows us to validate our level of understanding of the chemical interplay between metal elements and protein environments. The retrosynthetic construction, redesign, and directed evolution of artificial metalloglycosidases in this work demonstrated that proteinaceous space enables inorganic cofactors to accommodate thermodynamically and kinetically accessible reaction routes to activate glycosidic bonds. The successful conversion of OmpF into various metallohydrolases indicated that OmpF is apt for introducing an active site, reactions with multiple substrates, and whole-cell catalysis. Although membrane proteins have rarely been adapted for enzyme designs, our work demonstrated that OmpF could be a host macromolecule for various potential applications. Our work then leads to an intriguing question, why is there no precedence for such metal-dependent glycosidases in Nature. It might be related to the alternative metalloenzymes and relevant biomimetic complexes³⁹⁻⁴¹, which react with glycosides via Cu-dependent oxidative cleavage mechanism instead. The advantages of utilizing no metal or a redox-active metal element and dioxygen over a redox-inactive Zn ion can be further investigated. In addition, our work demonstrated that Zn-dependent and independent catalytic routes can co-emerge serendipitously. Such promiscuous reaction routes have been proposed when primordial or ancestor enzymes lead to divergent evolution. If then, artificially designed enzymes may resemble the states of nascent proteins in laboratory evolution and be employed as a surrogate to explore the evolutionary process.”

Q3-1) Will these artificial enzymes be chemically useful? What (extended) substrate scope do the variants have? Are the reported catalytic proficiencies superior to already existing tools? While I appreciate the up-scaling effort via click chemistry plus NMR characterisation (infos missing, see 4) I) below), the authors almost exclusively used highly reactive phenol-glucoside test substrates, which possess extremely good leaving groups. This feigns high catalytic activity - how does the reactivity towards “real” alkylglycosides (e.g. methyl glucoside) compare? I assume efficiencies will be much smaller...

We agree with the reviewer’s comment on the substrate. The substrates used in this work are highly activated ones, and the products of these reactions are unlikely to be useful for immediate applications. However, we aim to demonstrate that unprecedented metalloenzymes can emerge and evolve from a structure- and mechanism-based protein design rather than to develop biocatalysts for immediate application.

We indeed used highly activated substrates. However, they are commonly adapted substrates to monitor and characterize even natural enzymes due to the facile and efficient kinetic analysis.

In addition, as described in the revised manuscript, we found that OmpF2/E-R2 can react with octyl β -D-glucopyranoside, suggesting that we can expand the substrate scope. We are very interested in the evolution of OmpF variants to exhibit diverse substrate scope and better reactivity for more versatile applications in future studies.

Q3-2) Furthermore, they only compare the catalytic efficiency of the variants to their own published work of artificial beta-lactamases (line 98). In fact, the "catalytic proficiency of up to 2.8×10^9 " in the abstract only gets meaning when reading line 150 relating this value as an enhancement from the uncatalysed rate of glycosidase activity. The following references or others alike may be of use for this:

- Cowan J. A. et al., 2021, 10.1021/acs.inorgchem.0c01193
- Cowan J. A. et al., 2020, 10.1002/sml.202000392
- Nelson A.-G. D. et al., 2010, 10.1021/ic9014064
- Cowan J. A. et al., 2017, 10.1002/anie.201612079

I believe such information and wider perspective is highly necessary to be communicated to the scientific community in order for this manuscript to be of the high impact that Nature Communication embodies.

We sincerely appreciate the reviewer's comments. In the revised manuscript, we included the references to read, **"In addition, inorganic complexes, metallopolymers, and metallopeptides of showing glycosidase activities were reported⁹⁻¹²."**

"It might be related to the alternative metalloenzymes and relevant biomimetic complexes³⁹⁻⁴¹, which react with glycosides via Cu-dependent oxidative cleavage mechanism instead. The advantages of utilizing no metal or a redox-active metal element and dioxygen over a redox-inactive Zn ion can be further investigated."

We also revised the main text to compare the catalytic efficiency of OmpF variants to other artificial beta-lactamase as shown below, **"...Their catalytic efficiency or substrate-binding affinity were comparable or higher than those of Zn-complexes^{23,24} and artificial Zn-dependent β -lactamases,²⁰ although OmpF sequence was yet to be optimized."**

Q4) Furthermore, I highly recommend the authors to check the manuscript again and allow an English native speaker proofread the manuscript. There are many grammar / language mistakes or hyphenation inconsistencies, for example:

We appreciate the reviewer's comments. We revised the manuscript as advised below. Then, we also used the **Springer Nature editing service** as well to correct grammar and language mistakes.

a) whole cell & whole-cell (throughout the manuscript, e.g. page 3, introduction, line 47 and Figure 3 caption, line 595) should be "whole-cell" (when used as adjective)

We fixed the error in the revised manuscript.

b) present/ past tenses throughout the text, for example in lines 45-46 "This discrepancy leads us to question whether metal-dependent glycosidases can be synthesized" should be "led"

We fixed the error in the revised manuscript.

c) "All variants exhibited Zn-dependent esterase activities, demonstrating that catalytically active Zn-binding motifs are created within the outer membrane protein" should be "were created"

We fixed the error in the revised manuscript.

d) "The kinetic parameters of OmpF variants are similar, while their catalytic efficiency and substrate-binding affinity were substantially higher than those of artificial β -lactamases with analogous Zn sites." should be "were similar" just as "were substantially higher"

We fixed the error in the revised manuscript.

e) line 177: "supporting that the Zn site show sufficient basicity" should be "showed" (or "shows" if present tense was desired)

We fixed the error in the revised manuscript.

f) Repetitions do not read well. lines 57, 62 & 64: "Next", "Then", "Then"

We removed the redundant words in the revised manuscript.

g) line 66-67: "Both variants were obtained by E. coli heterologous expression" should be "heterologous expression in E. coli"

We fixed the error in the revised manuscript.

h) lines 192-194: "This work also provides a novel molecular basis to design and develop versatile biocatalysts to reengineer metabolic pathways and to produce biofuels with glycosides." comma missing after "biocatalysts"

We edited the sentence in the revised manuscript.

i) Article missing: "To bypass refolding step, we herein isolated" should be "To bypass the refolding step"

We added the article before "refolding step".

j) line 150: "uncatalysed rate of 4-methylumbelliferyl- β -D-glucopyranoside" a compound has no rate, do you mean the glycosidation?

We revised the sentence in the revised manuscript.

k) line 113-116: "To improve their basal activities of OmpF variants, we integrated the canonical Koshland mechanism that inverting and retaining β -glycosidases operate via a pair of acidic residues 6-11 Å and 5.1-5.5 Å apart, respectively" This sentence needs

revising.

We revised the sentence in the revised manuscript.

l) line 410: Did you measure the ^1H NMR at 400 MHz or 500 MHz? Both values are given. What product mass did you inject and what yield does it equal to?

We apologize for the error. We used 400 MHz ^1H NMR for the experiments. We used 0.3 mg of the reaction product (shown on top), and 1.5 mg of glucopyranosyl-4-phenyl-1H-1,2,3-triazole synthesized from 1-azido-1-deoxy-glucose (shown on bottom) for NMR experiments. Because we did not isolate 1-azido-1-deoxy-glucose from the reaction mixtures, which presumably include glucose and OG, the yield of the click reactions is unclear. However, reference 55 indicated that the Cu-catalyzed click reaction with 1-azido-1-deoxy-glucose shows 80–90% yield.

m) Figures: Usually, the figure captions come below the figure (in contrast to tables and their captions)

We moved the figure captions to come after the figure.

REVIEWER COMMENTS

Reviewer #1 (Remarks to the Author):

The revision improved the quality of this work very much. I have several comments about this.

1. Statistics of the crystal structures and figure presentations of them were significantly improved. The CC1/2 values indicate that higher resolution data can be included than the current cut-off. Generally, CC1/2 values around 0.6 in the highest resolution shell is allowed for resolution cut-off, if the completeness is high enough (>95% or so in the highest shell). But, it's up to the authors to do re-processing/refinement or not.
2. Additional data of the activities of Asp/Glu mutants (Supp. Figure 13) and pH-activity profile (Fig. 4d and Supp. Fig. 15) are very nice. Along with the LC/MS data (Fig. 5b), I am now convinced that the enzyme is highly likely a Koshland-type retaining glycoside hydrolase with the candidate nucleophile of Glu113 (D113E). Therefore, I think the rightmost reaction mechanism scheme in Figure 5c (inverting one with a water nucleophile) is not necessary anymore, and it is OK to indicate that Glu113 is a tentative nucleophile residue in that scheme.
3. "natural glycosidases show the pH optimum at pH 5.0–6.5". There are actually so many alkaliphilic glycosidases with pH optimum above pH 6.5. The acidic rim of the pH profile (~7.5 in Fig. 4d) may be a bit too high for a glutamate. However, some glycosidases have a high-pKa Glu in their active site, especially for general acid/base catalyst (For example, an alkaliphilic *Bacillus* GH11 xylanase. See PMID: 8756457). The basic rim of the pH profile (~8.7) should correspond to the initial general acid catalyst. This may be the Zn-coordinated water.

One more comment: Please add page and line numbers in the manuscript. Without those info, it is inconvenient for reviewers to designate a specific part.

Reviewer #2 (Remarks to the Author):

All of the most significant recommended changes, many of which required additional experiments, were made to the original manuscript. In particular, pH-dependent glycosidase activities studies have improved the understanding of the reaction mechanism. Therefore, this reviewer can now recommend publication in Nature Comm.

Reviewer #3 (Remarks to the Author):

The authors had previously submitted their work to Nature Communications end of 2021. Overall, the feedback of the three reviewers was mixed and critical, in order for this study to meet the high standards of Nature Communications and be accepted for publication.

The authors have significantly improved their manuscript considering all issues raised by reviewers. Most importantly, they have conducted further experiments (e.g. further crystallisation attempts, pH-dependency study ...). The revised manuscript has significantly improved in language and contents to a level sufficient to be publishable in the journal, essentially as is.

Reviewer #1

The revision improved the quality of this work very much. I have several comments about this.

Q1) Statistics of the crystal structures and figure presentations of them were significantly improved. The CC1/2 values indicate that higher resolution data can be included than the current cut-off. Generally, CC1/2 values around 0.6 in the highest resolution shell is allowed for resolution cut-off, if the completeness is high enough (>95% or so in the highest shell). But, it's up to the authors to do re-processing/refinement or not.

We sincerely appreciate the reviewer's comment. We previously used HKL2000, but we could not improve the datasets. Therefore, we rescaled the OmpF1 dataset using the XDS program instead, resulting in substantially enhanced higher resolution (from 3.35 to 3.10 Å) with comparable statistics (CC1/2 value of 0.839 and completeness of 98.1%), as shown below. Then, we conducted the data refinements of OmpF1 structure.

SUBSET OF INTENSITY DATA WITH SIGNAL/NOISE \geq -3.0 AS FUNCTION OF RESOLUTION _J													
RESOLUTION LIMIT	NUMBER OF REFLECTIONS			COMPLETENESS OF DATA	R-FACTOR observed	R-FACTOR COMPARED expected	I/SIGMA	R-meas	CC(1/2)	Anomal Corr _J	SigAno	Nano _J	
	OBSERVED	UNIQUE	POSSIBLE										
8.90	8150	2445	2616	93.5%	2.9%	2.9%	8098	35.31	3.4%	99.8*	30*	1.149	1124 _J
6.44	15372	4200	4231	99.3%	4.1%	3.9%	15369	26.33	4.8%	99.8*	13*	0.971	2042 _J
5.30	20038	5352	5391	99.3%	5.7%	5.7%	20035	18.77	6.7%	99.7*	3	0.846	2610 _J
4.61	23777	6305	6372	98.9%	6.6%	6.6%	23775	15.91	7.7%	99.7*	-1	0.796	3076 _J
4.14	26321	6992	7127	98.1%	8.6%	8.4%	26319	12.87	10.1%	99.5*	2	0.779	3398 _J
3.78	29449	7758	7955	97.5%	16.6%	16.9%	29447	7.03	19.4%	98.8*	1	0.712	3763 _J
3.51	32082	8445	8582	98.4%	25.1%	25.8%	32082	4.55	29.3%	98.7*	2	0.723	4125 _J
3.28	34936	9174	9248	99.2%	40.8%	42.0%	34936	3.01	47.6%	94.4*	0	0.688	4502 _J
3.10	36007	9678	9866	98.1%	77.7%	82.1%	35886	1.55	90.8%	83.9*	1	0.645	4684 _J
total	226132	60349	61388	98.3%	10.9%	11.0%	225947	10.33	12.7%	99.7*	3	0.762	29324 _J

1) before

2) after

Notably, the higher resolution data show much better B-factors as shown in Extended Data Table 1. Therefore, we revised the coordinate file of OmpF1, updated the validation report, and re-deposited it in the RCSB database. We also updated the figures in the main text (Figs 2a–b), Supplementary Figs 4 and 18, and Supplementary Table 3, using the newly processed OmpF1 dataset.

In addition, we revised the method section to read,

“X-ray diffraction data were processed and scaled using the program suite HKL2000⁵⁰ and XDS⁵¹. All data were processed using CCP4...”

We also revised the sentences to read,

“...resulting in a 2His/1Glu triad (2.0–2.4 Å for Zn-O/ ϵ N bonds)...”

“...Zn-bound exchangeable ligands as 7.9–9.5 Å and 4.7–5.1 Å in OmpF1 and OmpF2...”

Owing to the improved dataset, we can now assign the fourth ligand ligated to the Zn ion as a water molecule (B-factor: 54.84, 53.02, 68.14 for each protomer) with good agreement for the electron density maps. Thus, we slightly modified the sentence in the main text to read,

“...Zn-binding site. Regardless, the fourth site was ligated by a non-proteinaceous molecule, tentatively assigned as a water molecule. Consequently, OmpF1...”

Similarly, we re-processed the dataset of OmpF2 using both HKL2000 and XDS programs, as shown below. However, no improvement was observed, and we made no change to the OmpF2 figures/data.

Shell limit	Lower Angstrom	Upper Angstrom	Average I	Average error	Average stat.	Chi**2	Norm. R-fac	Linear R-fac	Square Rmeas	Rpim	CC1/2	CC*
30.00	8.62	1089.5	22.8	16.2	11.239	0.117	0.167	0.131	0.056	0.956	0.989	
8.62	6.87	535.7	9.8	7.1	7.362	0.104	0.137	0.113	0.042	0.991	0.998	
6.87	6.01	327.9	6.9	5.5	5.678	0.108	0.149	0.116	0.043	0.991	0.998	
6.01	5.46	262.4	6.2	5.1	5.238	0.121	0.190	0.131	0.049	0.977	0.994	
5.46	5.07	307.3	7.0	5.7	4.853	0.116	0.151	0.125	0.046	0.992	0.998	
5.07	4.78	283.9	6.9	5.7	4.789	0.125	0.156	0.134	0.050	0.984	0.996	
4.78	4.54	269.1	6.9	5.8	4.101	0.121	0.142	0.130	0.048	0.989	0.997	
4.54	4.34	238.1	6.7	5.7	3.722	0.124	0.141	0.133	0.049	0.991	0.998	
4.34	4.17	167.2	6.0	5.4	3.021	0.142	0.143	0.153	0.056	0.986	0.997	
4.17	4.03	127.1	5.7	5.3	2.876	0.182	0.182	0.195	0.072	0.988	0.997	
4.03	3.90	95.9	5.3	5.0	2.507	0.205	0.183	0.220	0.081	0.991	0.998	
3.90	3.79	81.2	5.2	5.0	2.371	0.239	0.185	0.258	0.094	0.989	0.997	
3.79	3.69	81.7	5.3	5.0	2.333	0.253	0.177	0.272	0.100	0.994	0.998	
3.69	3.60	66.3	5.3	5.1	2.207	0.303	0.161	0.326	0.120	0.996	0.999	
3.60	3.52	58.1	5.1	4.9	2.138	0.324	0.217	0.348	0.127	0.986	0.996	
3.52	3.45	45.7	5.0	4.9	2.006	0.383	0.230	0.411	0.150	0.985	0.996	
3.45	3.38	39.1	4.9	4.9	1.971	0.446	0.278	0.479	0.175	0.978	0.994	
3.38	3.31	30.4	4.9	4.8	1.917	0.553	0.331	0.594	0.217	0.962	0.990	
3.31	3.26	19.5	4.7	4.7	1.772	0.840	0.411	0.902	0.327	0.965	0.991	
3.26	3.20	19.1	4.7	4.7	1.714	0.819	0.474	0.879	0.319	0.944	0.985	
All reflections		206.6	6.8	5.8	3.607	0.150	0.161	0.163	0.061	0.939	0.984	

SUBSET OF INTENSITY DATA WITH SIGNAL/NOISE >= -3.0 AS FUNCTION OF RESOLUTION													
RESOLUTION LIMIT	NUMBER OF REFLECTIONS			COMPLETENESS OF DATA	R-FACTOR observed	R-FACTOR expected	COMPARED	I/SIGMA	R-meas	CC(1/2)	Anomal Corr	SigAno	Nano
8.91	7231	2422	2678	90.4%	7.8%	7.6%	6995	13.86	9.5%	98.3*	28*	1.121	973
6.45	14169	4347	4362	99.7%	8.0%	8.0%	14017	13.22	9.6%	98.5*	20*	0.990	1968
5.30	18315	5530	5539	99.8%	9.4%	9.0%	18177	11.36	11.2%	98.3*	14*	0.919	2564
4.61	22052	6572	6598	99.6%	10.0%	9.9%	21906	9.60	11.9%	98.4*	6	0.821	3080
4.14	24798	7370	7389	99.7%	11.5%	12.1%	24645	7.91	13.7%	98.6*	1	0.749	3468
3.78	27689	8197	8211	99.8%	18.8%	19.5%	27524	5.29	22.2%	98.1*	2	0.757	3863
3.50	30027	8871	8895	99.7%	25.0%	25.6%	29840	3.40	29.5%	98.9*	0	0.723	4187
3.28	32201	9496	9520	99.7%	39.9%	41.5%	32002	2.29	47.2%	96.4*	2	0.664	4485
3.10	33870	10122	10146	99.8%	76.5%	82.2%	33512	1.26	90.5%	90.8*	2	0.622	4682
total	210352	62927	63338	99.4%	12.9%	13.1%	208618	6.09	15.4%	98.7*	6	0.764	29270

Q2) Additional data of the activities of Asp/Glu mutants (Supp. Figure 13) and pH-activity profile (Fig. 4d and Supp. Fig. 15) are very nice. Along with the LC/MS data (Fig. 5b), I am now convinced that the enzyme is highly likely a Koshland-type retaining glycoside hydrolase with the candidate nucleophile of Glu113 (D113E). Therefore, I think the rightmost reaction mechanism scheme in Figure 5c (inverting one with a water nucleophile) is not necessary anymore, and it is OK to indicate that Glu113 is a tentative nucleophile residue in that scheme.

As suggested, we removed the inverting scheme and slightly modified the retaining mechanism in Figure 5c as shown below.

The main text is revised to read,

“For retaining β -glycosidases, such as OmpF2/E-R2, an acidic residue, possibly D113E, may initiate glycosylation as a nucleophile when Zn-OH₂ species acts as a Lewis acid. Then, the resulting Zn-OH may become a direct nucleophile or activate a water molecule as a Lewis base, facilitating the deglycosylation step.”

The figure legend is also revised to read,

“A proposed mechanism of Zn-dependent retaining β -glycosidase. The association of a water molecule is omitted for clarity.”

Q3) “natural glycosidases show the pH optimum at pH 5.0–6.5”. There are actually so many alkaliphilic glycosidases with pH optimum above pH 6.5. The acidic rim of the pH profile (~7.5 in Fig. 4d) may be a bit too high for a glutamate. However, some glycosidases have a high-pK_a Glu in their active site, especially for general acid/base catalyst (For example, an alkaliphilic *Bacillus* GH11 xylanase. See PMID: 8756457). The basic rim of the pH profile (~8.7) should correspond to the initial general acid catalyst. This may be the Zn-coordinated water.

We sincerely appreciate the reviewer’s comments. In the revised manuscript, we included the references about alkaliphilic *Bacillus* GH11 xylanase to read,

“...TONs at pH 8.0–8.5 and 7.5–8.0, respectively, whereas most natural glycosidases show the pH optimum at pH 5.0–6.5³⁰⁻³². But glycosidases, such as alkaline xylanases, exhibit catalytic reactivity at more basic conditions³³⁻³⁵. The pH-dependence of OmpF variants might be derived from...”

Q4) Please add page and line numbers in the manuscript. Without those info, it is inconvenient for reviewers to designate a specific part.

In the revised manuscript, we added page and line numbers as suggested.

Reviewer #2

All of the most significant recommended changes, many of which required additional experiments, were made to the original manuscript. In particular, pH-dependent glycosidase activities studies have improved the understanding of the reaction mechanism. Therefore, this reviewer can now recommend publication in Nature Comm. We appreciate the reviewer’s comments on our manuscript.

Reviewer #3

The authors had previously submitted their work to Nature Communications end of 2021. Overall, the feedback of the three reviewers was mixed and critical, in order for this study to meet the high standards of Nature Communications and be accepted for publication. The authors have significantly improved their manuscript considering all issues raised by reviewers. Most importantly, they have conducted further experiments (e.g. further crystallisation attempts, pH-dependency study ...). The revised manuscript has significantly improved in language and contents to a level sufficient to be publishable in the journal, essentially as is.

We appreciate the reviewer's comments on our manuscript.

In addition, we added the following sentences for clarity.

“The error bars shown in a, b, and e indicate the standard deviations of three runs of the experiments.”

“The error bars shown in a, c, and d indicate the standard deviations of three runs of the experiments”

“Coordinates and structure factors for the reported crystal structures in this work were deposited in the RCSB PDB under accession numbers 7FDY (OmpF1) and 7FF7 (OmpF2).”

REVIEWERS' COMMENTS

Reviewer #1 (Remarks to the Author):

The manuscript was revised without problems.